# PRO: Enabling P̲recise and R̲obust Text Watermark for O̲pen-Source LLMs

## Abstract

Text watermarking for large language models (LLMs) is important for model owners to verify the origin and protect the intellectual property of AI-generated text. While watermarking methods for closed-source LLMs' text generation are relatively mature, watermarking open-source LLMs' text generation remains challenging. Closed-source model developers typically embed text watermarks during decoding; however, this approach is ineffective for the text generation of open-source models, where developers have no control over how decoding occurs. As a result, owners of open-source LLMs still lack practical methods to verify whether a given piece of AI-generated text originated from their models. The primary challenge lies in embedding watermarks directly into model weights without compromising detection accuracy. One possible solution is first to create a text generation watermark in the closed-source setting, then distill that watermark information into the publicly released model's weights. However, this approach faces two critical challenges: (i) Reduced detectability due to inconsistency between the watermark patterns learned by the model and the predefined patterns used during detection. This inconsistency arises because existing closed-source watermark patterns are difficult for models to learn effectively. (ii) Vulnerability to modifications by downstream users, such as fine-tuning or model merging, which may weaken or completely remove the embedded watermark. To address these challenges, we propose **PRO**, a precise and robust text watermarking method for open-source LLMs. First, we introduce a trainable watermark policy model, which is jointly optimized with the LLM during training. This co-optimization helps generate watermark patterns that are easier for the model to learn, significantly reducing inconsistencies between generated patterns and predefined detection criteria. Additionally, we incorporate a regularization term into the watermarking loss, which simulates various perturbations (e.g., fine-tuning, model merging) and penalizes any degradation in watermark detectability under these modifications. This approach ensures that the embedded watermark remains resilient even after downstream model alterations. Our evaluation on mainstream open-source LLMs (e.g., LLaMA-3.2, LLaMA-3, and Phi-2) demonstrates that our approach significantly outperforms prior methods in terms of both watermark detectability and robustness against model modifications.

## 1 Introduction

With the rapid advancement of LLMs and their widespread deployment, researchers and regulators have raised growing concerns regarding their potential misuse in generating harmful content (Bommasani et al., 2021; Union, 2021). To address this issue, text watermarking has emerged as a promising technique that embeds a watermark signal during text generation to facilitate the detection of LLM-generated content (Hastuti et al., 2025; Ghanim et al., 2025). Mainstream approaches (Kuditipudi et al., 2024; Kirchenbauer et al., 2023; Hu et al., 2023) leverage a watermark scheme $f_w$ to generate watermark logits that bias the decoding process. During detection, the statistical distribution of tokens in the text is analyzed using $f_w$ to determine if it has been watermarked. As illustrated in Figure 1 (a), these decoding-based methods assume a closed-source LLM setting, where the LLM owner controls the entire inference pipeline, including the integration of $f_w$ into the decoding process.

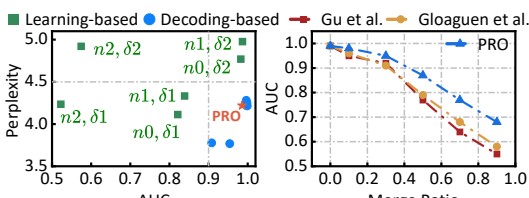

Figure 1: Text watermarking for (a) closed-source and (b) open-source LLMs. Closed-source watermarking relies on watermark decoding, while open-source watermarking requires embedding the watermark into the model weights so that standard decoding still produces watermarked text.

As open-source LLMs rapidly improve, the performance gap between open-source and closed-source LLMs is narrowing. Notably, some open-source LLMs such as DeepSeek (Liu et al., 2024c) and LLaMA-4 (Meta, 2024) are now matching or even surpassing closed-source LLMs like OpenAI's GPT-4 and Claude 3.5 on specific benchmarks (Guo et al., 2024). This underscores an urgent need for effective text watermarking for open-source LLMs. However, decoding-based methods are fundamentally unsuitable in the open-source setting, where LLM users will have full access to the inference pipeline to remove the watermark decoding. As illustrated in Figure 1 (b), a viable alternative involves embedding watermarking capability directly into the LLM's weights, enabling the LLM to generate watermarked texts naturally without relying on watermark decoding. In this context, two key challenges arise: (i) the *detectability* of watermarks in generated text, and (ii) the *robustness* against users' modifications on LLM's weights.

To integrate watermarking mechanisms directly into LLM weights, Christ et al. (2024) proposed shifting the addition of watermark logits from the decoding phase to a direct modification of the bias terms in the final projection layer. However, since prominent open-source LLMs typically omit bias terms in this layer, this requires architectural modifications that users could easily detect and remove. An alternative and promising direction, which avoids such architectural alterations, is to train the LLM on watermarked text (Gu et al., 2024; Sander et al., 2024). The objective here is for the LLM to natively learn the statistical patterns of the watermark, thereby generating watermarked logits as an inherent part of its output.

Despite its promise, existing learning-based watermarking faces critical challenges in *learnability* and *robustness*. First, unlike decoding-based watermarking, where the watermark logits are manually injected during the decoding process, learning-based methods require the LLM to learn to generate watermarked text directly. Its detectability is therefore highly contingent on the learnability of watermark signals, typically requiring a large watermark logits magnitude $\delta$ during training. However, training on highly distorted text impairs the LLM's generation quality. As shown in Figure 2 (left), learning-based watermark (■) trained with $\delta = 1$ only achieves

Figure 2: *(Left)* learning-based and decoding-based KGW watermarks under varying watermark hyperparameters $n$ and $\delta$. *(Right)* existing learning-based watermarks' AUC after merging with the unwatermarked LLM.

0.84 AUC, while achieving 0.99 AUC with $\delta = 2$ results in significant degradation of generation quality, i.e., Perplexity (PPL) increases from 4.1 to 5.0. This contrasts with decoding-based methods (●) that achieve both high quality (PPL $< 4.5$) and strong detectability (AUC $> 0.9$) simultaneously. Additionally, current learning-based methods are limited to small $n$-gram lengths ($n \leq 1$) (Gu et al., 2024; Sander et al., 2024; Zhao et al., 2025), as larger $n$ values significantly increase the complexity of watermark patterns for LLMs to learn, thereby diminishing watermark learnability. Even with $\delta = 2$, the LLMs fail to learn 2-gram KGW watermarks (AUC $= 0.54$). However, practical applications require higher $n$-gram (e.g., 3- or 4-grams) for robustness against reverse engineering (Kirchenbauer et al., 2023; Zhao et al., 2025). Furthermore, the open-source setting allows users to modify LLMs through techniques like fine-tuning or model merging, which can inadvertently or intentionally remove learned watermarks. As shown in Figure 2 (right), merging a watermarked LLM with an

unwatermarked one at a 0.5 ratio reduces the watermark detection AUC to $0.79$. In contrast, the proposed PRO improves robustness, achieving an AUC of $0.87$, as reported in Table1 in Section 4.

In this paper, we propose **PRO** (**P**recise and **R**obust **O**pen-source LLM Watermark). Our journey begins by analyzing the aforementioned issues of existing watermarks for open-source LLMs.

First, current open-source LLM watermarking methods (Gu et al., 2024; Sander et al., 2024) distill predefined watermark patterns into model weights and use the same patterns for detection. This creates a *Generation-Detection Inconsistency*: the LLM often learns a deviated version of the watermark, while detection assumes the original pattern. This inconsistency stems from simply adopting watermarking methods like KGW (Kirchenbauer et al., 2023) that were originally built for closed-source LLMs and were not intentionally designed to be learnable. These methods use predefined mapping functions to generate watermark logits. From the LLM's perspective, these mappings often appear arbitrary, forcing the LLM to memorize fragmented associations between prefixes and watermark logits rather than internalizing a coherent pattern of watermark logits. This challenge becomes catastrophic for large $n$-gram watermarks, where the complexity of mapping functions renders learning difficult. To resolve this, **PRO** introduces a trainable watermark policy that dynamically optimizes the watermark mapping function through joint training with the LLM. This co-adaptation ensures the policy generates watermark patterns that can be effectively learned by LLMs. Crucially, during detection, **PRO** employs the optimized policy instead of the predefined one, ensuring alignment with the watermark patterns the LLM has actually learned.

Second, to enhance robustness against user's weight modifications, Gloaguen et al. (2025) proposed embedding watermarks into "stable" parameters identified by observing value changes after fine-tuning on natural text. However, as shown in Figure 2 (right), this approach offers only marginal robustness against fine-tuning, since parameter stability is inherently dataset-dependent, and modifications like model merging or fine-tuning on other datasets can still erase watermarks. To address this, we propose the concept of *forgotten perturbation*: perturbations to weights that maximally degrade watermark detectability. To attenuate its negative impact, **PRO** iteratively generates perturbations using anti-watermarked text (adversarially crafted to erase watermarks), simulating powerful *forgotten perturbation*. During training, the LLM learns to withstand the *forgotten perturbation* by minimizing its disruptive effects while maintaining watermark detectability. By unifying perturbation resistance with watermark training in a single optimization framework, **PRO** achieves robustness against diverse user modifications.

We evaluate **PRO** on three mainstream open-source LLMs, including LLaMA3-8B, LLaMA3.2-3B (AI@Meta, 2024) and Phi2-2.7B (Microsoft, 2023). We embed watermarks and assess their robustness under common user modifications, such as quantization, pruning, fine-tuning, and model merging. Results reveal that **PRO** can achieve high detectability, low quality degradation and large $n$-gram lengths (i.e., $n \geq 5$), as shown in Figure 2 (left). These results represent substantial improvements over state-of-the-art open-source watermarking (Gu et al., 2024; Gloaguen et al., 2025) and can even match the performance of closed-source counterpart. Additionally, watermarks embedded by **PRO** are more robust against user modifications. **PRO** consistently preserves high detectability (AUC $\geq 0.80$) under aggressive model modifications, including high-ratio merging and long-step fine-tuning, marking the first precise and robust text watermark for open-source LLMs.

## 2 RELATED WORK AND THREAT MODEL

### 2.1 TEXT WATERMARKS FOR CLOSED-SOURCE LLMS

To ensure traceability of content generated by LLMs, watermarking techniques have been proposed to embed identifiable statistical signals into model outputs. Among these, *decoding-based watermarking* (Kirchenbauer et al., 2023; Christ et al., 2024; Zhao et al., 2024; Kirchenbauer et al., 2024) is a widely adopted approach. The watermark decoding function $f_w(\pi_\theta(x), \xi)$ leverages a secret key $\xi$ to transform the original next-token distribution $\pi_\theta(\cdot \mid x)$ into a modified distribution that embeds a detectable watermark signal in the generated text. This enables post-hoc detection via a test function $f_d(x, \xi)$, which computes a p-value indicating the presence of a watermark. However, decoding-based watermarking relies on customized decoding algorithms and is not applicable in open-model settings where users control the decoding process.

## 2.2 TEXT WATERMARKS FOR OPEN-SOURCE LLMs

Current watermarking schemes for open-source language models can be broadly categorized into two types: *Input Prompt-dependent* and *Input Prompt-independent*. The former requires access to the input prompt for detection, while the latter can detect watermarks using only the output text. Notably, our **PRO** falls into the category without requiring the input prompt.

### 2.2.1 INPUT PROMPT-DEPENDENT

Several recent works propose open-source watermarking techniques that require input prompt for detection. Xu et al. (2025) jointly train a detector and an LLM to embed detectable signals into the output, requiring both the prompt and output for detection. Block et al. (2025) perturb model weights with Gaussian noise and detect watermarks via gradient-based statistical tests, which also rely on the input prompt. Both methods require access to the input prompt during detection, limiting their applicability in real-world scenarios.

### 2.2.2 INPUT PROMPT-INDEPENDENT

We further divide input prompt-independent watermarking methods into two classes: learning-based watermarks and structural-editing watermarks.

*Learning-based Watermark.* Gu et al. (2024) shows that decoding-based watermarks can be embedded into model weights by distilling the watermark from a teacher LLM $\pi_o$. Then, the same watermark scheme can be used to detect the watermark in the student LLM $\pi_\theta$. The teacher generates watermarked data $\mathcal{D}_{\text{wm}}$, which the student fine-tunes using the cross-entropy loss:

$$\mathcal{L}_{\text{sampling}}(\pi_\theta) = \mathbb{E}_{x \sim \mathcal{D}_{\text{wm}}} \left[ \sum_{t=1}^{|x|} - \log \pi_\theta(x_t | x_{<t}) \right] \quad (1)$$

Alternatively, by exploiting the logits, the student can be fine-tuned to mimic the teacher LLM's next-token distribution using the KL-divergence loss:

$$\mathcal{L}_{\text{logit}}(\pi_\theta) = \mathbb{E}_{x \sim \mathcal{D}} \left[ \sum_{t=1}^{|x|} \text{KL}(f_w(\pi_o(\cdot \,|\, x_{<t}), \xi_w) \,\|\, \pi_\theta(\cdot \,|\, x_{<t})) \right] \quad (2)$$

*Structural-editing Watermarks.* Christ et al. (2024) embed watermarks by adding small, token-specific Gaussian perturbations to the output-layer bias vector of the model. To detect whether a text sequence is watermarked, the LLM owner aggregates the bias perturbations of each output token. As most open-source LLMs disable output-layer biases by default[1], implementing this method requires explicitly enabling the bias term, which introduces an architectural modification. Such changes remain identifiable and can be easily removed by analyzing and modifying the architecture of the model.

## 2.3 THREAT MODEL

We consider a threat model where an open-source LLM is publicly released to users, who have full access to the model's weights and architecture. Users may modify the model through fine-tuning, model merging, quantization, or pruning. The LLM owner embeds a watermark into the released model's weights and retains a private detection mechanism, which may be made available via a detection API. Detection is performed solely on the generated text, without access to the user's input prompt or control over the decoding process.

## 3 METHODS

### 3.1 OVERVIEW

Learning-based watermarking aims to train LLMs to generate watermarked text natively, without modifying the decoding scheme. Specifically, given an original LLM $\pi_o$ and a watermarked decoding

---

[1]For example, the output layers of LLaMA, Qwen, and Mistral models in the Hugging Face Transformers library are defined with `bias=False`.

function $f_w$, the combination acts as a teacher model. The objective is to train a student LLM $\pi_\theta$ with standard decoding such that its next-token distribution $\pi_\theta(x)$ approximates the teacher's output, $f_w(\pi_o(x), \xi)$, for any input $x$. Our goal is to build a precise and robust learning-based watermark for open-source LLMs that (i) maintains high detection accuracy without degrading generation quality, and (ii) remains robust against general user modifications such as fine-tuning. We first discuss the core challenges in achieving these goals and then present how our proposed **PRO** addresses them.

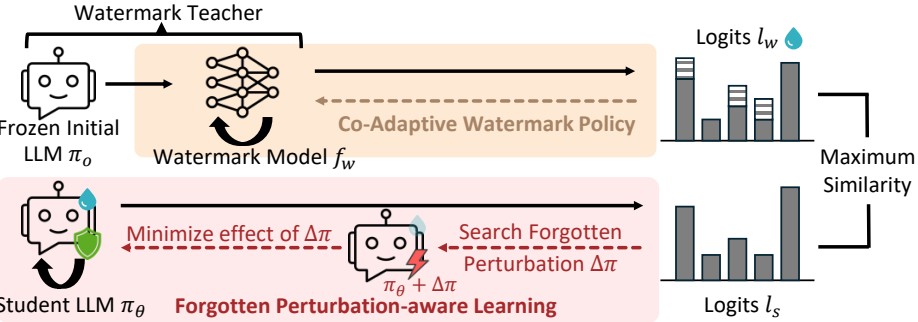

Figure 3: Overview of our proposed method. CAWP (*upper*) jointly trains a watermark model with the student LLM to generate learning-friendly watermark logits. FPL (*bottom*) improves robustness by searching and minimizing the effect of forgotten perturbations that may erase the watermark.

## 3.2 CO-ADAPTIVE WATERMARK POLICY (CAWP)

Watermark detection for open-source LLMs involves two steps: (1) the LLM user generates text via the watermarked LLM, and (2) the LLM owner uses the predefined watermark pattern to detect the text. In this case, an important consideration is the *consistency* between the watermark pattern learned by the LLM and the predefined one used during detection. To assess this *consistency*, we fine-tune a LLaMA3-8B on text generated by a decoding-based watermarked LLaMA3-8B (i.e., watermark teacher), thereby obtaining a learning-based watermarked LLaMA3-8B (i.e., watermark student). We then measure and compare the green token ratios across them in Figure 4. The results reveal a significant green ratio drift in the student relative to the teacher, which reduces the AUC from $0.99$ to $0.84$, indicating that the student LLM fails to fully internalize the watermarking pattern of the watermark teacher. To mitigate the inconsistency, we identify two primary optimization directions. The first is to design a learning-friendly watermark pattern, thereby enhancing its inherent learnability for the student LLM. The second is to adapt the detection mechanism itself, using a pattern that aligns more closely with the watermark distribution actually learned by the student LLM, rather than strictly adhering to the original predefined pattern.

Unlike prior work that uses rigid, predefined watermark functions, we introduce Co-Adaptive Watermark Policy (CAWP) as illustrated in Figure 3 (upper). It co-optimizes a trainable watermark policy model with the student LLM, allowing the watermark patterns to adapt to the LLM's learning dynamics and become more learnable. Crucially, detection leverages the co-optimized patterns rather than the original predefined ones, ensuring detection aligns with the watermark signals internalized by the LLM to mitigate generation-detection inconsistency.

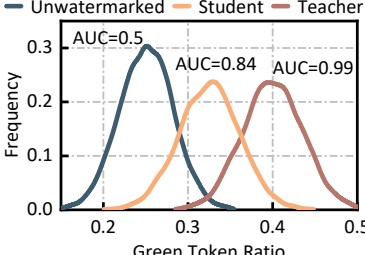

Figure 4: Green token ratios for unwatermarked, watermark student, and watermark teacher (KGW with green ratio $= 0.25$, $n = 1$, $\delta = 1$).

The watermark policy consists of a pre-trained Embedding Model $E$ (e.g., BERT (Devlin et al., 2019)) and a trainable Watermark Mapping Model $M$ (e.g., an MLP). Given a sequence of preceding tokens $\{x_{i-n:i-1}\}$ at position $i$, $E$ generates their embeddings $\{e_{i-n:i-1}\}$. These embeddings are then transformed by $M$ into raw watermark logits over the vocabulary, i.e., $M(E(x_{i-n:i-1}))$. The final watermark logits are obtained by scaling with a strength coefficient $\delta$, yielding $\delta \cdot M(E(x_{i-n:i-1}))$, which are added to the next-token logits before sampling. Given a dataset of texts $\mathcal{D}$, the training objective is to minimize the mean KL divergence between teacher and student next token distributions

on all prefixes in $\mathcal{D}$:

$$\mathcal{L}_{\text{sim}} = \frac{1}{|\mathcal{D}|} \sum_{x \in \mathcal{D}} \sum_{i=n}^{N-1} \text{KL}\left[ \overbrace{P\Big(\log\big(\pi_o(\cdot \mid x_{\leq i})\big) + \underbrace{\delta \cdot M(E(x_{i-n:i-1}))}_{\text{watermark logits}}\Big)}^{\text{teacher distribution}} \;\Big\|\; \overbrace{\pi_\theta(\cdot \mid x_{\leq i})}^{\text{student distribution}} \right] \quad (3)$$

Here, $n$ is the gram length and $P(\cdot)$ converts the logits to probabilities. Both the teacher LLM $\pi_o$ and the embedding model $E$ are frozen; only the student model $\pi_\theta$ and mapping model $M$ are optimized. The student LLM $\pi_\theta$ is optimized by minimizing $\mathcal{L}_{\text{sim}}$, while the mapping model $M$ must also satisfy the following requirements: (i) *Unbiased Token Preference*: the watermark logits should not exhibit persistent positive or negative bias toward specific tokens. (ii) *Balanced Watermark Logits*: the logits should have zero mean across the vocabulary, ensuring symmetric perturbation that makes approximately half the tokens more likely and the other half less likely. (iii) *Non-Vanishing Watermark Logits*: to avoid degenerate solutions where the watermark logits vanish, each watermark logit is encouraged to maintain a minimum absolute magnitude, regularized toward a target value $\epsilon$.

$$\mathcal{L}_{\text{norm}} = \overbrace{\sum_i \left| \frac{1}{|\mathcal{V}|} \sum_j M(e_i)^{(j)} \right|}^{\text{Balanced Watermark Logits}} + \overbrace{\sum_j \left| \frac{1}{N} \sum_i M(e_i)^{(j)} \right|}^{\text{Unbiased Token Preference}} + \lambda_1 \overbrace{\sum_{i,j} \max\left(0, \epsilon - |M(e_i)^{(j)}|\right)}^{\text{Non-Vanishing Watermark Logits}} \quad (4)$$

$M(e_i)^{(j)}$ denotes the watermark logit for the $j$-th token in the vocabulary (of size $|\mathcal{V}|$) corresponding to the $i$-th input embedding $e_i$, with $N$ total input samples. The index $i$ sums over all samples, and $j$ sums over all vocabulary tokens. The final training loss for the watermark mapping model $M$ is the weighted sum of the similarity loss and the normalization loss, given by:

$$\mathcal{L}_M = \mathcal{L}_{\text{sim}} + \lambda_2 \mathcal{L}_{\text{norm}} \quad (5)$$

Upon training, the watermarked student LLM $\pi_\theta$ is released, and the co-optimized watermark mapping model $M$ is retained for detection. For a given text, the LLM owner computes the watermark logit at each position $i$ by first embedding the $n$-gram prefix with $E$, then transforming it via the mapping model $M$ to obtain the watermark logit for the actual next token $x_i$. The detection score is the average watermark logit across the sequence, given by $z = 1/N \sum_{i=1}^{N} M\big(E(x_{i-n:i-1})\big)^{(x_i)}$. If $z$ exceeds a predefined threshold, the text is considered watermarked.

While some prior work also employs neural networks to generate watermark logits, they target different goals. For example, Liu et al. (2024b), Ren et al. (2023) and Liu et al. (2024a) design semantic-invariant or public-verifiable watermark models to improve robustness or enable third-party verification. In contrast, our CAWP framework is fundamentally different. Existing approaches are designed for closed-source LLMs and do not consider the learnability of the watermark by the model itself—their watermark models are trained independently from the LLM. By contrast, CAWP focuses on generating *learning-friendly* watermark logits by jointly optimizing the watermark model with the student LLM being watermarked. Additionally, such joint training ensures a bidirectional alignment: the watermark pattern used for detection adapts toward the student LLM's learned representation, while the student LLM is simultaneously guided to internalize patterns consistent with detection. This mutual convergence narrows the gap between the student and teacher distributions in Figure 4.

### 3.3 FORGOTTEN PERTURBATION-AWARE LEARNING (FPL)

The vulnerability of watermarked LLMs to user modifications arises primarily from *forgotten perturbation*. Specifically, when a user modifies a watermarked LLM $\pi_\theta$ into $\pi'_\theta$, the weight drift $\Delta\pi_\theta = \pi'_\theta - \pi_\theta$ may remove the learned watermark. We term the drift that erases the watermark as *forgotten perturbation*, and our goal is to minimize its effect during the watermark learning stage.

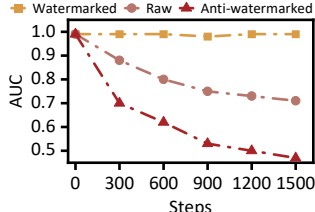

Figure 5: AUC during fine-tuning on raw, watermarked, and anti-watermarked texts.

To understand its root cause, we investigate fine-tuning as a representative user modification, i.e., $\pi_{t+1} = \pi_t - \eta\, g(\pi_t)$, where $g(\pi_t)$ is the gradient on user data driving weight drift. Figure 5 shows that fine-tuning a watermarked LLaMA3-8B on raw texts (a mixture of green/red tokens) collapses detectability, whereas fine-tuning on watermarked texts (green-token

dominant) preserves it. This indicates that gradients arising from red tokens are the primary cause of *forgotten perturbation*. To further validate this, we fine-tune on *anti-watermarked* texts, generated with inverted watermark logits to make texts red-token dominant. As shown in Figure 5, fine-tuning on these anti-watermarked texts results in a substantial drop in detectability. These results confirm that red-token-driven *forgotten perturbations* are the dominant factor in watermark forgetting, and mitigating their impact should be a key objective.

To achieve this, we propose Forgotten Perturbation-aware Learning (FPL), which explicitly reduces model sensitivity to *forgotten perturbation*, as illustrated in Figure 3 (bottom). Our objective is to train a watermarked model that can not only generate watermarked texts but also attenuate the effect of future perturbations caused by red-token updates. The training objective of the LLM is defined as:

$$
\arg\min_{\boldsymbol{\pi}_\theta} \mathcal{L}_{\mathrm{sim}}(\boldsymbol{\pi}_\theta) + \beta \left( \mathcal{L}_{\mathrm{anti}}(\boldsymbol{\pi}_\theta) - \mathcal{L}_{\mathrm{anti}}(\boldsymbol{\pi}_\theta - \alpha \frac{\nabla \mathcal{L}_{\mathrm{anti}}(\boldsymbol{\pi}_\theta)}{\|\nabla \mathcal{L}_{\mathrm{anti}}(\boldsymbol{\pi}_\theta)\|}) \right) \tag{6}
$$

Here, $\mathcal{L}_{\mathrm{sim}}$ is the watermark learning loss over watermarked texts (as defined in Equation 3), and $\mathcal{L}_{\mathrm{anti}}$ is the watermark forgetting loss evaluated on anti-watermarked texts, defined analogous to $\mathcal{L}_{\mathrm{sim}}$ using KL divergence loss. The second term measures the change in forgetting loss before and after applying a normalized step in the direction of the *forgotten perturbation*. The regularization weight $\beta$ controls the trade-off, and $\alpha$ is the perturbation step size. This formulation encourages the model to not only learn a strong watermark but also be robust against potential forgetting induced by user-side modifications.

Although the optimization in Equation 6 involves second-order derivatives, we can solve it using three forward/backward passes: (1) evaluate $\mathcal{L}_{\mathrm{sim}}(\pi_\theta)$ on watermarked logits, (2) compute $\mathcal{L}_{\mathrm{anti}}(\pi_\theta)$ on anti-watermarked data and perform a backward pass to obtain $\nabla \mathcal{L}_{\mathrm{anti}}(\pi_\theta)$, and (3) simulate one normalized fine-tuning step along the forgetting gradient and perform another forward pass to compute $\mathcal{L}_{\mathrm{anti}}(\pi_\theta - \alpha \hat{g})$, where $\hat{g} = \nabla \mathcal{L}_{\mathrm{anti}} / \|\nabla \mathcal{L}_{\mathrm{anti}}\|$. The difference between the pre- and post-step losses reflects the watermarked LLM's vulnerability to *forgotten perturbation*, which we aim to minimize. It is worth noting that Equation 6 is used only for training the LLM $\pi_\theta$, while the watermark mapping model $M$ is still optimized with the loss in Equation 5. For completeness, the formal convergence analysis of CAWP is presented in the Appendix B, along with the detailed mathematical derivation of the second-order derivative term in Equation 6.

# 4 EXPERIMENTS

## 4.1 EXPERIMENTAL SETTINGS

**Models.** We perform experiments on three open-source LLMs: LLaMA3-8B, Phi2-2.7B, and LLaMA3.2-3B. These models cover a range of model sizes from lightweight to large-scale models. The watermark mapping model $M$ is a lightweight MLP: a linear projection $\mathbb{R}^{1024} \to \mathbb{R}^{500}$, two ReLU-based residual blocks, and a final projection to the vocabulary dimension with $\tanh$ to constrain logits to $[-1, 1]$.

**Implementation Details.** In our watermarking pipeline, the semantic embeddings are generated by `compositional-bert-large` (Chanchani & Huang, 2023). During training, both the student LLM $\pi_\theta$ and the watermark mapping model $M$ are jointly optimized to maximize learnability and detection consistency. We set the regularization weights $\lambda_1 = \lambda_2 = 1$ in the training loss for CAWP. For the FPL component, we use perturbation step size $\alpha = 0.1$ and regularization weight $\beta = 5$. In our ablation study, we further analyze the sensitivity of watermark robustness to variations in these hyperparameters.

**Baseline Watermarking Methods.** Our experiments include three baseline watermarking methods, all implemented via KL-based distillation from a decoding-based teacher model. The first is Gu et al.-KGW. Following the setup in (Gu et al., 2024), we distill KGW with a fixed $\gamma = 0.25$ and evaluate three configurations of $(k, \delta)$: $(1, 2)$, $(0, 1)$, and $(1, 1)$. The second baseline is Gu et al.-KTH, which distills the exponential decoding watermarking scheme proposed in (Kuditipudi et al., 2024). The third is Gloaguen et al.-KGW, which builds on KGW by embedding watermarks into "stable" parameters identified via contrastive task vector analysis before and after fine-tuning on raw data. The details of training configurations and device usage can be found in Appendix F.3.

**Metrics.** We evaluate open-source watermarking methods along three dimensions: detectability, text quality, and robustness. For detectability, we follow (Gu et al., 2024), where each watermarked model generates 5,000 samples by prompting with 50-token prefixes from the C4 dataset (Dodge et al., 2021) and generating 200-token continuations. Standard sampling with temperature 1 is used during generation to ensure consistency across methods. Detection is performed by comparing these generations against 5,000 non-watermarked texts using the corresponding watermark detection algorithm, and we report the Area Under the ROC Curve (AUC) and True Postive Rate under different False Positive Rate (TPR@FPR). Text quality is measured via median perplexity (PPL), using a LLaMA-2-13B model. To assess robustness, we apply four types of model modifications that simulate real-world user behavior, modification settings are in Appendix F.3.

### 4.2 RESULTS

### 4.2.1 COMPARISON WITH EXISTING WORKS

**Detectability and Quality Analysis.** To demonstrate **PRO**'s effectiveness, we compare it against existing text watermarking methods for open-source LLMs, including (Gu et al., 2024) and (Gloaguen et al., 2025). To assess the tradeoff between detectability and generation quality, we vary the watermark hyperparameters ($\delta$ and $n$-gram for KGW, $s$ for KTH, $\delta$ for **PRO**) and measure perplexity and detection TPR at 1% FPR (TPR@1). The hyperparameter configuration can be found in the Appendix F.1. As shown in Figure 6, existing open-source watermarks exhibit quality degradation, while **PRO** shows the lowest effect on the quality of text. **PRO** bridges this gap: by co-optimizing a dynamic watermark policy with the watermarked LLM, it discovers learning-friendly watermark patterns that LLMs can internalize under low-distortion conditions. This achieves a TPR@1 of 0.92 while reducing perplexity by $20.5\%$ compared to the best baseline across all tested LLMs. The results in terms of AUC and TPR at other 0.1% and 10% FPR can be found in Figure 8 and Table 8. ROC curves are shown in Figure 7.

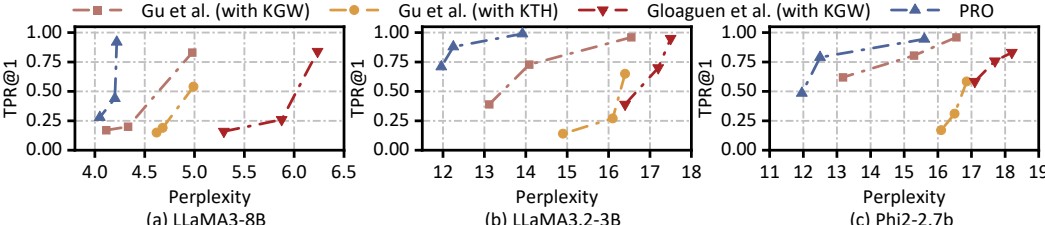

Figure 6: Effectiveness comparison of different open-source LLMs text watermarks, in terms of detection TPR at 1% FPR (TPR@1) and median PPL on three LLMs. A better watermark detectability and generation quality is indicated by higher TPR@1 and lower PPL, as shown by lines closer to the upper left corner. The original models' PPL are: LLaMA3-8B = 3.6, LLaMA3.2-3B = 10.9, Phi2-2.7B = 11.3.

**Robustness against Model Modifications.** Following the evaluation setup of (Gloaguen et al., 2025), we evaluate the robustness of **PRO** under common model modification scenarios, including quantization, pruning, model merging, and fine-tuning. As shown in Table 1, existing watermarking methods generally perform reliably under *quantization* and *pruning*, where the parameters shifts to the model are limited and the impact on learned weights is relatively small. However, under *model merging* and *fine-tuning*, which induce more substantial and less predictable changes in LLM weights, the performance of existing methods degrades noticeably. For example, when merging a watermarked LLM with the original one one at an interpolation ratio of $t = 0.5$, the TPR@5FPR of (Gu et al., 2024) with KTH decreases to 0.09, while **PRO** retains a higher 0.57. For fine-tuning, after 300 fine-tuning steps on the OpenMathInstruct dataset, the TPR@1FPR of the (Gu et al., 2024) with KTH drops to 0.08, whereas **PRO** maintains 0.43. Consistent improvement is also shown on OpenCodeInstruct dataset. These results indicate that **PRO** exhibits improved resilience to more aggressive forms of model modifications. We attribute this improvement to the incorporation of FPL, which explicitly simulates and counteracts watermark forgetting via forgotten perturbations during training.

**Computational Efficiency.** Our method maintains comparable computational cost to prior watermarking approaches. CAWP introduces only a lightweight MLP watermark model (1.16M parameters, 0.0388% of a 3B LLM), which is negligible relative to the base model. FPL requires two additional

forward/backward passes per iteration, but this overhead is offset by faster convergence: PRO reaches AUC 0.997 within 2000 steps, whereas KGW requires 5000 steps for a maximum AUC of 0.991. As a result, the overall wall-clock training time remains comparable to that of the baselines (see Appendix C for detailed analysis).

Table 1: TPR at 5%, 1% FPR and PPL of different watermark methods of LLaMA-3-8B under model modifications. The perplexity of the original LLaMA-3-8B without watermarking is 3.6.

| Category | Method | Config | Gu et al. (2024)-KTH | | | Gu et al. (2024)-KGW | | | Gloaguen et al. (2025)-KGW | | | PRO | | |
|---|---|---|---|---|---|---|---|---|---|---|---|---|---|---|
| | | | TPR@5 | TPR@1 | PPL | TPR@5 | TPR@1 | PPL | TPR@5 | TPR@1 | PPL | TPR@5 | TPR@1 | PPL |
| Unaltered | | | 0.78 | 0.54 | 5.0 | 0.96 | 0.83 | 4.9 | 0.94 | 0.84 | 6.2 | 0.99 | 0.92 | 4.2 |
| Quantization | 8 bits | GPTQ | 0.66 | 0.42 | 5.2 | 0.92 | 0.79 | 5.0 | 0.95 | 0.84 | 6.3 | 0.97 | 0.86 | 4.3 |
| | | INT8 | 0.69 | 0.45 | 5.1 | 0.91 | 0.73 | 5.0 | 0.94 | 0.84 | 6.4 | 0.93 | 0.77 | 4.4 |
| | 4 bits | HQQ | 0.54 | 0.29 | 6.1 | 0.95 | 0.83 | 5.5 | 0.91 | 0.77 | 7.9 | 0.98 | 0.91 | 4.8 |
| | | GPTQ | 0.64 | 0.36 | 5.5 | 0.91 | 0.74 | 5.1 | 0.94 | 0.81 | 7.3 | 0.94 | 0.79 | 5.1 |
| Pruning | Wanda | $\rho = 0.2$ | 0.73 | 0.47 | 8.6 | 0.96 | 0.84 | 7.5 | 0.97 | 0.90 | 9.9 | 0.99 | 0.92 | 7.0 |
| | | $\rho = 0.5$ | 0.64 | 0.35 | 8.4 | 0.86 | 0.66 | 6.9 | 0.93 | 0.80 | 8.9 | 0.96 | 0.84 | 6.2 |
| | SparseGPT | $\rho = 0.2$ | 0.78 | 0.55 | 7.7 | 0.95 | 0.82 | 8.1 | 0.95 | 0.84 | 9.0 | 0.98 | 0.91 | 7.1 |
| | | $\rho = 0.5$ | 0.76 | 0.50 | 8.1 | 0.91 | 0.74 | 7.8 | 0.93 | 0.78 | 9.6 | 0.97 | 0.89 | 8.0 |
| Merging | SLERP | $t = 0.1$ | 0.36 | 0.15 | 5.2 | 0.76 | 0.53 | 4.3 | 0.82 | 0.59 | 6.1 | 0.92 | 0.78 | 4.2 |
| | | $t = 0.3$ | 0.21 | 0.08 | 5.1 | 0.65 | 0.39 | 4.0 | 0.62 | 0.37 | 5.3 | 0.83 | 0.60 | 4.1 |
| | | $t = 0.5$ | 0.09 | 0.02 | 4.8 | 0.30 | 0.12 | 3.8 | 0.34 | 0.14 | 4.6 | 0.57 | 0.28 | 4.0 |
| | | $t = 0.7$ | 0.06 | 0.02 | 4.3 | 0.15 | 0.05 | 3.7 | 0.19 | 0.06 | 4.2 | 0.31 | 0.12 | 3.9 |
| | | $t = 0.9$ | 0.06 | 0.01 | 4.1 | 0.11 | 0.03 | 3.6 | 0.11 | 0.03 | 4.0 | 0.17 | 0.06 | 3.8 |
| Finetuning | OpenMath Instruct | $s = 300$ | 0.21 | 0.08 | 5.1 | 0.48 | 0.20 | 4.1 | 0.45 | 0.22 | 5.5 | 0.70 | 0.43 | 4.1 |
| | | $s = 600$ | 0.17 | 0.05 | 4.9 | 0.38 | 0.16 | 3.7 | 0.41 | 0.19 | 5.2 | 0.51 | 0.26 | 3.9 |
| | | $s = 900$ | 0.09 | 0.03 | 4.8 | 0.30 | 0.11 | 3.6 | 0.31 | 0.13 | 5.2 | 0.46 | 0.22 | 3.7 |
| | | $s = 1200$ | 0.09 | 0.02 | 4.7 | 0.25 | 0.09 | 3.5 | 0.25 | 0.10 | 5.0 | 0.40 | 0.18 | 3.6 |
| | | $s = 1500$ | 0.02 | 0.02 | 4.5 | 0.23 | 0.08 | 3.5 | 0.22 | 0.08 | 5.1 | 0.37 | 0.17 | 3.6 |
| | OpenCode Instruct | $s = 300$ | 0.27 | 0.11 | 7.3 | 0.63 | 0.40 | 6.9 | 0.65 | 0.41 | 7.4 | 0.71 | 0.43 | 7.0 |
| | | $s = 600$ | 0.22 | 0.08 | 5.7 | 0.45 | 0.24 | 4.9 | 0.48 | 0.23 | 7.0 | 0.56 | 0.31 | 5.4 |
| | | $s = 900$ | 0.16 | 0.06 | 4.8 | 0.32 | 0.14 | 5.0 | 0.34 | 0.16 | 6.8 | 0.48 | 0.25 | 5.1 |
| | | $s = 1200$ | 0.11 | 0.03 | 4.7 | 0.29 | 0.12 | 4.3 | 0.31 | 0.12 | 6.3 | 0.40 | 0.18 | 4.8 |
| | | $s = 1500$ | 0.07 | 0.02 | 4.6 | 0.26 | 0.10 | 4.2 | 0.30 | 0.12 | 5.9 | 0.39 | 0.17 | 4.0 |

**Robustness against Paraphrasing attack.** We evaluate the robustness of our watermarking method under paraphrasing using DIPPER(Krishna et al., 2023), a controllable paraphraser that rewrites text while preserving semantics in Table 2. We use two settings: DIPPER-1, with lexical diversity set to 60 and order diversity 0; and DIPPER-2, with lexical diversity 60 and order diversity 20. Our method shows consistent detection performance under both settings, indicating robustness against paraphrase attacks.

### 4.2.2 DISTILLATION DATA SCALING STUDY

Recent work by Gloaguen et al. (2025) shows that watermark robustness can be improved by substantially enlarging the distillation dataset. In Table 1, PRO uses a 64 million tokens distillation corpus to maintain the trade-off between watermark robustness and model fidelity. To examine how PRO benefits from data scaling, we conduct

Table 2: AUC under DIPPER attacks.

| Watermark Method | DIPPER1 | DIPPER2 |
|---|---|---|
| (Gu et al., 2024)-KTH | 0.82 | 0.79 |
| (Gu et al., 2024)-KGW | 0.86 | 0.84 |
| (Gloaguen et al., 2025)-KGW | 0.80 | 0.74 |
| **PRO (Ours)** | **0.90** | **0.87** |

additional experiments where the distillation corpus is expanded from 32M to 160M tokens. Table 3 summarizes the results under two robustness settings: (1) model merging (ratio=0.5) and (2) 1500-step fine-tuning on OpenMathInstruct (Toshniwal et al., 2024). We observe consistent improvements in TPR@5% as the distillation data increases. For example, TPR@5% FPR improves from 0.49 to 0.69 under merge-0.5, and from 0.29 to 0.72 under fine-tuning. The perplexity increases only moderately (4.0 to 5.3), indicating that data scaling does not substantially harm model quality.

### 4.2.3 ABLATION STUDY

To understand the contributions of each component proposed, we conduct an ablation study by varying the perturbation step size $\alpha$ and the regularization weight $\beta$ in Equation 6. When both are set to zero, the model essentially reduces to using CAWP alone. In this case, robustness improves slightly over the baseline (Merge AUC = 0.82 after merging), indicating CAWP alone can already

| Modification | 32M | 64M | 96M | 128M | 160M |
|---|---|---|---|---|---|
| Merge-0.5 | 0.49 | 0.57 | 0.63 | 0.67 | 0.69 |
| FT-1500 | 0.29 | 0.37 | 0.50 | 0.63 | 0.72 |
| PPL | 4.0 | 4.2 | 4.7 | 5.1 | 5.3 |

Table 3: Effect of distillation data scaling on watermark robustness. Larger distillation datasets improve TPR@5%FPR under both merge-0.5 and FT-1500 settings, with only moderate PPL increase.

enhance watermark robustness. Introducing FPL via non-zero $\beta$ brings further gains in robustness (AUC = 0.87 at $\alpha = 0.1$, $\beta = 5$), though with a mild increase in perplexity. However, larger values of $\alpha$ or $\beta$ lead to degraded or unstable results, highlighting the importance of careful hyperparameter tuning to balance robustness and generation quality.

Table 4: Ablation on perturbation step size $\alpha$ and regularization weight $\beta$. We report AUC and PPL of unaltered model and AUC after merging ($t = 0.5$) to indicate robustness.

| $\alpha$ | 0 | **0.1** | 1 | 2 | 5 |
|---|---|---|---|---|---|
| AUC | 0.99 | **0.99** | 0.98 | 0.97 | 0.95 |
| PPL | 4.18 | **4.22** | 4.30 | 4.70 | 5.00 |
| Merge AUC | 0.82 | **0.87** | 0.86 | 0.83 | 0.79 |

| $\beta$ | 0 | 1 | **5** | 10 |
|---|---|---|---|---|
| AUC | 0.99 | 1.00 | **0.99** | 0.95 |
| PPL | 4.18 | 4.21 | **4.22** | 4.52 |
| Merge AUC | 0.82 | 0.85 | **0.87** | 0.84 |

(a) Effect of perturbation step size $\alpha$.     (b) Effect of regularization weight $\beta$.

### 4.2.4 OTHER EXPERIMENTS

We provide additional results in the Appendix, including convergence analysis, computational efficiency, robustness under model modifications, and evaluations on extra datasets and models. We also report TPRs at multiple FPRs, robustness to paraphrasing, and comparisons with classifier-based detectors. These supplemental experiments further demonstrate that PRO generalizes well across models, datasets, and evaluation settings.

## 5 CONCLUSION

We identify the key challenges in watermarking open-source LLMs, the low learnability of predefined watermark patterns and their vulnerability to model modifications. To address these issues, we propose **PRO**, a precise and robust framework that jointly trains a learnable watermark policy with the LLM and incorporates perturbation-aware optimization to enhance robustness. **PRO** improves watermark detectability and resilience while maintaining generation quality, providing a practical solution for open-source LLM text watermark.

## 6 ETHICS STATEMENT

This work focuses on developing robust watermarking techniques for open-source large language models to support provenance verification and responsible AI use. Our study does not involve human or animal subjects, nor does it require collection of personal or sensitive data. All experiments are conducted on publicly available pretrained models and benchmark datasets. We believe our method enhances transparency and accountability in AI deployment, and we are not aware of any ethical concerns or potential harms arising from this research.

## 7 REPRODUCIBILITY STATEMENT

We have taken multiple steps to ensure the reproducibility of our work. The code is publicly available at https://anonymous.4open.science/r/PRO, together with a README file that includes instructions for installation, configuration, and execution of experiments.

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

APPENDIX

## A    THE USE OF LARGE LANGUAGE MODELS (LLMS)

The authors used ChatGPT and Grammarly to check and correct any typos and grammatical errors.

## B    THEORETICAL ANALYSIS

### B.1    CONVERGENCE ANALYSIS OF CAWP

We anchor our proof in established convergence guarantees for distillation-based LLM training, where a student LLM $\pi_\theta$ converges to a fixed teacher $\pi_o$ under KL divergence minimization. The core innovation of CAWP is introducing a trainable watermark policy $M_\phi$ that perturbs the teacher distribution. We prove convergence by analyzing this perturbation.

#### B.1.1    BASELINE DISTILLATION CONVERGENCE (FIXED POLICY $M$)

For a fixed watermark policy $M_{\text{fix}}$, the loss reduces to standard distillation:

$$\mathcal{L}_{\text{fix}}(\theta) = \text{KL}\left(\pi_o + \delta M_{\text{fix}} \parallel \pi_\theta\right). \tag{7}$$

Under Lipschitz continuity and bounded gradients, gradient descent on $\theta$ converges:

$$\lim_{t \to \infty} \|\nabla_\theta \mathcal{L}_{\text{fix}}\| = 0. \tag{8}$$

This serves as the baseline we extend.

#### B.1.2    CAWP'S JOINT OPTIMIZATION

CAWP introduces a trainable policy $M_\phi$, leading to the joint loss:

$$\mathcal{L}_{\text{sim}}(\theta, \phi) = \underbrace{\text{KL}\left(\pi_o + \delta M_\phi \parallel \pi_\theta\right)}_{\text{Distillation loss}} + \lambda_2 \underbrace{\mathcal{L}_{\text{norm}}(\phi)}_{\text{Regularizer}}. \tag{9}$$

We analyze the effect of $M_\phi$ on convergence. Specifically, $\mathcal{L}_{\text{sim}}(\theta, \phi)$ satisfies the following assumptions:

- **Smoothness:** $\nabla \text{KL}(\cdot \parallel \pi_\theta)$ is $L_\theta$-Lipschitz in $\theta$, and $\nabla M_\phi$ is $L_\phi$-Lipschitz in $\phi$. LLMs and MLPs with smooth activations (e.g., GELU, Tanh) satisfy this.
- **Boundedness:** $\|M_\phi\| \leq B_M$ and $\|\nabla_\phi M_\phi\| \leq G_M$, since outputs are bounded by the last Tanh layer and gradients are bounded via clipping.
- **Convexity:** $\mathcal{L}_{\text{norm}}$ is convex in $\phi$, as $\ell_1$ penalties in Eq. (4) enforce convexity.

Given these assumptions, alternating gradient descent on $\theta$ and $\phi$ converges to a stationary point, where $M_\phi$ learns watermark mappings that perturb $\pi_o$ without degrading the LLM's performance.

#### B.1.3    ALTERNATING GRADIENT DESCENT DYNAMICS

In CAWP, optimization alternates between updating $\theta$ (distilling to the perturbed teacher) and $\phi$ (adapting the watermark policy):

$$\theta^{t+1} = \theta^t - \eta_\theta \nabla_\theta \mathcal{L}_{\text{sim}}(\theta^t, \phi^t), \tag{10}$$

$$\phi^{t+1} = \phi^t - \eta_\phi \nabla_\phi \mathcal{L}_{\text{sim}}(\theta^{t+1}, \phi^t). \tag{11}$$

The distillation term encourages $\pi_\theta$ to match the perturbed distribution, while $\mathcal{L}_{\text{norm}}(\phi)$ (e.g., $\ell_1$ on policy outputs) prevents $M_\phi$ from over-perturbing, ensuring watermark detectability without degrading text quality.

Under the smoothness assumption, the loss is $L$-smooth overall ($L = L_\theta + \delta L_\phi B_M$). Hence,

$$
\mathcal{L}_{\text{sim}}(\theta^{t+1}, \phi^{t+1}) \leq \mathcal{L}_{\text{sim}}(\theta^t, \phi^t) - \left( \frac{\eta_\theta}{2} \|\nabla_\theta \mathcal{L}_{\text{sim}}\|^2 + \frac{\eta_\phi}{2} \|\nabla_\phi \mathcal{L}_{\text{sim}}\|^2 \right)
$$
$$
+ \frac{L\eta_\theta^2}{2} \|\nabla_\theta \mathcal{L}_{\text{sim}}\|^2 + \frac{L\eta_\phi^2}{2} \|\nabla_\phi \mathcal{L}_{\text{sim}}\|^2.
\tag{12}
$$

Choosing $\eta_\theta, \eta_\phi \leq 1/L$ ensures monotonic decrease:

$$
\mathcal{L}_{\text{sim}}(\theta^{t+1}, \phi^{t+1}) \leq \mathcal{L}_{\text{sim}}(\theta^t, \phi^t) - c\Big( \|\nabla_\theta \mathcal{L}_{\text{sim}}\|^2 + \|\nabla_\phi \mathcal{L}_{\text{sim}}\|^2 \Big), \quad c > 0.
\tag{13}
$$

### B.1.4 CONVERGENCE TO STATIONARY POINT

Summing the descent inequality over $T$ iterations gives:

$$
\sum_{t=0}^{T-1} \left( \|\nabla_\theta \mathcal{L}_{\text{sim}}(\theta^t, \phi^t)\|^2 + \|\nabla_\phi \mathcal{L}_{\text{sim}}(\theta^t, \phi^t)\|^2 \right) \leq \frac{\mathcal{L}_{\text{sim}}(\theta^0, \phi^0) - \mathcal{L}_{\text{sim}}^*}{cT},
\tag{14}
$$

where $\mathcal{L}_{\text{sim}}^*$ is the infimum of the loss. As $T \to \infty$, the gradients vanish:

$$
\lim_{t \to \infty} \|\nabla_\theta \mathcal{L}_{\text{sim}}\| = 0, \quad \lim_{t \to \infty} \|\nabla_\phi \mathcal{L}_{\text{sim}}\| = 0.
\tag{15}
$$

The boundedness of $M_\phi$ and its gradients ensures that the perturbation $\delta M_\phi$ remains controlled, preventing divergence. Moreover, the $\mu$-strong convexity of $\mathcal{L}_{\text{norm}}$ implies that, for fixed $\theta$, the subproblem in $\phi$ is strongly convex, guaranteeing convergence to a unique minimizer $\phi^*(\theta)$ that balances watermark strength and regularization.

### B.2 FORMAL DETECTABILITY ADVANTAGE OF CAWP OVER FIXED WATERMARK MAPPINGS

In this section, we formally show that CAWP achieves a strictly smaller generation-detection inconsistency compared to the baseline (fixed watermark mappings). Our analysis leverages (i) the irreducible non-approximability of hash-like fixed mappings and (ii) standard generalization bounds for the joint hypothesis class of student LLMs and adaptive watermark policies.

**Notation.** Let $\pi_\theta(y \mid x)$ denote the student LLM, and let $q_{\text{fix}}(y \mid x)$ denote the teacher distribution generated by a fixed watermark mapping $M_{\text{fix}}$ (such as KGW/KTH). For CAWP, let $q_\phi(y \mid x)$ denote the teacher distribution generated by the trainable watermark policy $M_\phi$.

Define the population KL risks:

$$
\mathcal{R}_{\text{fix}}(\theta) = \mathbb{E}_x \, \text{KL}\big(q_{\text{fix}}(\cdot \mid x) \,\|\, \pi_\theta(\cdot \mid x)\big),
\tag{16}
$$
$$
\mathcal{R}_{\text{CAWP}}(\theta, \phi) = \mathbb{E}_x \, \text{KL}\big(q_\phi(\cdot \mid x) \,\|\, \pi_\theta(\cdot \mid x)\big).
\tag{17}
$$

The optimal risks for the two settings are:

$$
\gamma := \inf_\theta \mathcal{R}_{\text{fix}}(\theta), \qquad \varepsilon := \inf_{\theta, \phi} \mathcal{R}_{\text{CAWP}}(\theta, \phi).
\tag{18}
$$

**Assumption 1 (Irreducible error of baseline fixed mappings).** The fixed watermark mapping $M_{\text{fix}}$ is hash-like, discontinuous, and not aligned with the transformer parameterization. Thus, there exists a constant $\gamma_0 > 0$ such that

$$
\forall \theta : \quad \mathcal{R}_{\text{fix}}(\theta) \geq \gamma_0,
\tag{19}
$$

and hence:

$$
\gamma = \inf_\theta \mathcal{R}_{\text{fix}}(\theta) \geq \gamma_0.
\tag{20}
$$

This corresponds to the empirical mismatch observed in Figure 4, where the student fails to reproduce the fixed green-token distribution.

**Assumption 2 (Generalization of CAWP).** Let $\widehat{\mathcal{R}}_{\text{CAWP}}(\theta, \phi)$ denote the empirical CAWP loss over $N$ training samples:

$$\widehat{\mathcal{R}}_{\text{CAWP}}(\theta, \phi) = \frac{1}{N} \sum_{i=1}^{N} \text{KL}\big(q_\phi(\cdot \mid x_i) \| \pi_\theta(\cdot \mid x_i)\big). \tag{21}$$

We assume that the joint hypothesis class $\mathcal{H} = \{(\theta, \phi)\}$ has finite capacity (e.g., finite Rademacher complexity), following standard generalization analyses of deep neural networks (Bartlett & Mendelson, 2002; Neyshabur et al., 2015; Bartlett et al., 2017). We do not require the explicit value of $\mathfrak{C}$, only that it is finite, which holds for bounded-parameter networks such as the CAWP policy and the student LLM. Moreover, the KL loss is bounded for all softmax-based language models, as both $q_\phi(\cdot \mid x)$ and $\pi_\theta(\cdot \mid x)$ are strictly positive distributions over a finite vocabulary.

Under these standard conditions, classical uniform-convergence bounds (Bartlett & Mendelson, 2002) imply that there exists a constant $C > 0$ such that, with probability at least $1 - \delta$,

$$\mathcal{R}_{\text{CAWP}}(\theta, \phi) \leq \widehat{\mathcal{R}}_{\text{CAWP}}(\theta, \phi) + C \sqrt{\frac{\mathfrak{C} + \log(1/\delta)}{N}}. \tag{22}$$

**Assumption 3 (Effective optimization of CAWP).** Let $(\theta^\star, \phi^\star)$ be an approximate empirical minimizer:

$$\widehat{\mathcal{R}}_{\text{CAWP}}(\theta^\star, \phi^\star) \leq \inf_{\theta, \phi} \widehat{\mathcal{R}}_{\text{CAWP}}(\theta, \phi) + \eta, \tag{23}$$

where $\eta$ is negligible. Because $M_\phi$ adapts to the representational capacity of the student LLM, we may assume:

$$\inf_{\theta, \phi} \widehat{\mathcal{R}}_{\text{CAWP}}(\theta, \phi) \approx 0, \qquad \widehat{\mathcal{R}}_{\text{CAWP}}(\theta^\star, \phi^\star) \approx 0. \tag{24}$$

**Theorem B.1** (Detectability gain of CAWP)**.** *Under Assumptions 1–3, with probability at least $1 - \delta$, the optimal risks satisfy*

$$\gamma \geq \gamma_0 \quad and \quad \varepsilon \leq C \sqrt{\frac{\mathfrak{C} + \log(1/\delta)}{N}}. \tag{25}$$

*Consequently, the relative and absolute gaps between CAWP and fixed watermark mappings are bounded by*

$$\frac{\varepsilon}{\gamma} \leq \frac{C}{\gamma_0} \sqrt{\frac{\mathfrak{C} + \log(1/\delta)}{N}} \quad and \quad \gamma - \varepsilon \geq \gamma_0 - C \sqrt{\frac{\mathfrak{C} + \log(1/\delta)}{N}}. \tag{26}$$

The proof is detailed:

*Step 1: Lower bound on $\gamma$.* Immediate from Assumption 1.

*Step 2: Upper bound on $\varepsilon$.* Applying the generalization bound Equation 22 to $(\theta^\star, \phi^\star)$ gives

$$\mathcal{R}_{\text{CAWP}}(\theta^\star, \phi^\star) \leq \widehat{\mathcal{R}}_{\text{CAWP}}(\theta^\star, \phi^\star) + C \sqrt{\frac{\mathfrak{C} + \log(1/\delta)}{N}}. \tag{27}$$

Using Assumption 3 yields

$$\mathcal{R}_{\text{CAWP}}(\theta^\star, \phi^\star) \leq C \sqrt{\frac{\mathfrak{C} + \log(1/\delta)}{N}}. \tag{28}$$

Since $\varepsilon$ is the infimum over all $(\theta, \phi)$,

$$\varepsilon \leq \mathcal{R}_{\text{CAWP}}(\theta^\star, \phi^\star) \leq C \sqrt{\frac{\mathfrak{C} + \log(1/\delta)}{N}}. \tag{29}$$

*Step 3: Relative and absolute gaps.* Combining $\gamma \geq \gamma_0$ and the above yields

$$\frac{\varepsilon}{\gamma} \leq \frac{C}{\gamma_0} \sqrt{\frac{\mathfrak{C} + \log(1/\delta)}{N}}. \tag{30}$$

Furthermore,

$$\gamma - \varepsilon \geq \gamma_0 - C\sqrt{\frac{\mathfrak{C} + \log(1/\delta)}{N}}. \tag{31}$$

This completes the proof.

**Discussion.** The bound above shows that the detectability gain of CAWP over fixed watermark mappings is driven by three intuitive factors: (i) a larger irreducible error $\gamma_0$ of the fixed mapping (harder-to-learn hash-like patterns), (ii) a larger training sample size $N$, which shrinks the generalization term, and (iii) a smaller effective capacity $\mathfrak{C}$ of the CAWP model. In other words, CAWP yields the largest improvement precisely in the regime where fixed mappings are intrinsically hard to learn, while CAWP is trained with sufficient data and a well-regularized architecture.

### B.3  FORMAL ROBUSTNESS ADVANTAGE OF FPL

**Robustness setting.** Let $\theta$ denote the parameters of a watermarked student model, and let $\theta' = \theta + \Delta$ be a user-modified model after some fine-tuning, merging, or other downstream adaptation. We assume the user perturbation is bounded in parameter space, $\|\Delta\| \leq R$, where $R > 0$ is a given radius (e.g., induced by model merging or a limited number of fine-tuning steps).

Let $q_{\text{anti}}(\cdot \mid x)$ denote the *anti-watermark teacher* distribution, obtained by subtracting the green-list bias from the original watermark logits (cf. Eq. (6) in the main text). We define the *anti-watermark loss*

$$L_{\text{anti}}(\theta) := \mathbb{E}_{x \sim D_{\text{anti}}}\Big[\text{KL}\big(q_{\text{anti}}(\cdot \mid x) \,\|\, \pi_\theta(\cdot \mid x)\big)\Big], \tag{32}$$

where $D_{\text{anti}}$ is a distribution of "anti-watermark" inputs (e.g., adversarial prompts or samples generated using the anti-watermark logits). A smaller value of $L_{\text{anti}}(\theta)$ means that the student model behaves more like the anti-watermark teacher on these hard inputs, i.e., the embedded watermark is weaker. Thus "watermark forgetting" corresponds to a *decrease* in $L_{\text{anti}}$.

For a given radius $R$, we therefore measure robustness by the worst-case *drop* of $L_{\text{anti}}$ under bounded perturbations:

$$\Delta_{\max}^{\downarrow}(\theta; R) := \max_{\|\Delta\| \leq R} \Big\{ L_{\text{anti}}(\theta) - L_{\text{anti}}(\theta + \Delta) \Big\}. \tag{33}$$

A model with smaller $\Delta_{\max}^{\downarrow}(\theta; R)$ is more robust, because no admissible user modification can significantly push the model towards the anti-watermark distribution.

**Assumption 4 (Smoothness of the anti-watermark loss).** $L_{\text{anti}}(\theta)$ is $L$-smooth, i.e., its gradient is Lipschitz continuous with constant $L > 0$:

$$\big\|\nabla L_{\text{anti}}(\theta_1) - \nabla L_{\text{anti}}(\theta_2)\big\| \leq L \|\theta_1 - \theta_2\| \quad \forall\, \theta_1, \theta_2. \tag{34}$$

This is a standard assumption in optimization for deep networks.

**Assumption 5 (FPL reduces the forgetting gradient).** Let $\theta_{\text{base}}$ be the parameters obtained *without* FPL (e.g., using CAWP or a baseline watermarking scheme), and let $\theta_{\text{FPL}}$ be the parameters obtained *with* FPL. Define

$$G_{\text{base}} := \big\|\nabla L_{\text{anti}}(\theta_{\text{base}})\big\|, \qquad G_{\text{FPL}} := \big\|\nabla L_{\text{anti}}(\theta_{\text{FPL}})\big\|. \tag{35}$$

Since FPL explicitly penalizes moving the model towards the anti-watermark teacher along the worst forgetting direction (see Eq. (6)), it encourages solutions where the anti-watermark loss is locally flat. We therefore assume that for some $\kappa \in (0, 1)$ we have

$$G_{\text{FPL}} \leq \kappa\, G_{\text{base}}, \tag{36}$$

i.e., FPL reduces the norm of the anti-watermark gradient by a multiplicative factor $\kappa$ compared to the baseline. In Section X.X we empirically verify this effect by measuring $\|\nabla L_{\text{anti}}(\theta)\|$ for models trained with and without FPL.

**Theorem B.2** (Robustness gain of FPL under bounded perturbations)**.** *Under Assumptions 4–5 and for any perturbation radius $R > 0$, the worst-case drops of the anti-watermark loss for the baseline*

*model and the FPL-enhanced model satisfy*

$$\Delta_{\max}^{\downarrow}(\theta_{\text{base}}; R) \ \leq \ R\,G_{\text{base}} + \frac{L}{2}R^2, \tag{37}$$

$$\Delta_{\max}^{\downarrow}(\theta_{\text{FPL}}; R) \ \leq \ R\,G_{\text{FPL}} + \frac{L}{2}R^2 \ \leq \ R\,\kappa G_{\text{base}} + \frac{L}{2}R^2. \tag{38}$$

*Consequently, FPL reduces the worst-case watermark forgetting by at least*

$$\Delta_{\max}^{\downarrow}(\theta_{\text{base}}; R) - \Delta_{\max}^{\downarrow}(\theta_{\text{FPL}}; R) \ \geq \ R(1 - \kappa)\,G_{\text{base}}. \tag{39}$$

*In particular, for any fixed radius R, smaller $\kappa$ (stronger FPL regularization) and larger baseline forgetting gradient $G_{\text{base}}$ lead to a larger robustness gain.*

*Proof.* By $L$-smoothness of $L_{\text{anti}}$, we have for any $\theta$ and any perturbation $\Delta$ the standard lower bound

$$L_{\text{anti}}(\theta + \Delta) \geq L_{\text{anti}}(\theta) + \nabla L_{\text{anti}}(\theta)^{\top}\Delta - \frac{L}{2}\|\Delta\|^2. \tag{40}$$

Rearranging terms gives

$$L_{\text{anti}}(\theta) - L_{\text{anti}}(\theta + \Delta) \leq -\nabla L_{\text{anti}}(\theta)^{\top}\Delta + \frac{L}{2}\|\Delta\|^2. \tag{41}$$

Maximizing the right-hand side over all perturbations with $\|\Delta\| \leq R$ yields (by aligning $\Delta$ with the gradient):

$$\max_{\|\Delta\| \leq R} \left\{ L_{\text{anti}}(\theta) - L_{\text{anti}}(\theta + \Delta) \right\} \leq R\left\|\nabla L_{\text{anti}}(\theta)\right\| + \frac{L}{2}R^2. \tag{42}$$

By the definition of $\Delta_{\max}^{\downarrow}(\theta; R)$, this implies

$$\Delta_{\max}^{\downarrow}(\theta; R) \leq R\left\|\nabla L_{\text{anti}}(\theta)\right\| + \frac{L}{2}R^2. \tag{43}$$

Applying this bound to $\theta_{\text{base}}$ and $\theta_{\text{FPL}}$ gives

$$\Delta_{\max}^{\downarrow}(\theta_{\text{base}}; R) \leq R\,G_{\text{base}} + \frac{L}{2}R^2, \tag{44}$$

$$\Delta_{\max}^{\downarrow}(\theta_{\text{FPL}}; R) \leq R\,G_{\text{FPL}} + \frac{L}{2}R^2. \tag{45}$$

Using Assumption 5 ($G_{\text{FPL}} \leq \kappa G_{\text{base}}$) yields

$$\Delta_{\max}^{\downarrow}(\theta_{\text{FPL}}; R) \leq R\,\kappa G_{\text{base}} + \frac{L}{2}R^2. \tag{46}$$

Therefore,

$$\Delta_{\max}^{\downarrow}(\theta_{\text{base}}; R) - \Delta_{\max}^{\downarrow}(\theta_{\text{FPL}}; R) \geq \left( R\,G_{\text{base}} + \frac{L}{2}R^2 \right) - \left( R\,\kappa G_{\text{base}} + \frac{L}{2}R^2 \right) \tag{47}$$

$$= R(1 - \kappa)\,G_{\text{base}}. \tag{48}$$

This shows that FPL strictly reduces the worst-case *drop* of the anti-watermark loss under any bounded perturbation $\|\Delta\| \leq R$, with the robustness gain scaling linearly in both $(1 - \kappa)$ and $G_{\text{base}}$. $\qquad\square$

## B.4 COMPUTATIONAL OVERHEAD OF FPL

In this section, we provide additional details on the computational overhead of the proposed Forgotten Perturbation-aware Learning (FPL). Although the optimization objective in Equation 6 involves second-order derivatives, we show that it can be solved efficiently without computing exact Hessian information.

$$\arg\min_{\pi_\theta} \mathcal{L}_{\text{sim}}(\pi_\theta) + \beta \left( \mathcal{L}_{\text{anti}}(\pi_\theta) - \mathcal{L}_{\text{anti}}\left( \pi_\theta - \alpha\frac{\nabla\mathcal{L}_{\text{anti}}(\pi_\theta)}{\|\nabla\mathcal{L}_{\text{anti}}(\pi_\theta)\|} \right) \right) \tag{49}$$

where $\mathcal{L}_{\text{anti}}(\cdot)$ denotes the forgetting loss on anti-watermarked texts. Intuitively, the second term measures the decrease of $\mathcal{L}_{\text{anti}}$ after one normalized fine-tuning step along the forgetting gradient. By minimizing this gap, the model not only learns a strong watermark but also remains robust to potential forgetting induced by user-side modifications.

To solve this perturbation minimization problem, we consider an iterative gradient method (e.g., SGD). By the chain rule, the update rule is:

$$\pi_\theta^{t+1} = \pi_\theta^t - \eta \left( \nabla\mathcal{L}_{\text{sim}}(\pi_\theta^t) + \beta \left( \nabla\mathcal{L}_{\text{anti}}(\pi_\theta^t) - \nabla\mathcal{L}_{\text{anti}}\left( \pi_\theta^t - \alpha\frac{\nabla\mathcal{L}_{\text{anti}}(\pi_\theta^t)}{\|\nabla\mathcal{L}_{\text{anti}}(\pi_\theta^t)\|} \right) \underbrace{\nabla\left( \pi_\theta^t - \alpha\frac{\nabla\mathcal{L}_{\text{anti}}(\pi_\theta^t)}{\|\nabla\mathcal{L}_{\text{anti}}(\pi_\theta^t)\|} \right)}_{\text{second-order term}} \right) \right) \tag{50}$$

where $\eta$ is the learning rate. The last factor involves a second-order term (i.e., Hessian information), which is expensive to compute. Following prior work (Finn et al., 2017; Rajeswaran et al., 2019), we approximate this second-order term as a constant. The update rule then simplifies to:

$$\pi_\theta^{t+1} = \pi_\theta^t - \eta \left( \nabla\mathcal{L}_{\text{sim}}(\pi_\theta^t) + \beta \left( \nabla\mathcal{L}_{\text{anti}}(\pi_\theta^t) - \nabla\mathcal{L}_{\text{anti}}\left( \pi_\theta^t - \alpha\frac{\nabla\mathcal{L}_{\text{anti}}(\pi_\theta^t)}{\|\nabla\mathcal{L}_{\text{anti}}(\pi_\theta^t)\|} \right) \right) \right) \tag{51}$$

With this approximation, FPL requires only three forward/backward passes per optimization step:

1. A forward pass to compute $\mathcal{L}_{\text{sim}}(\pi_\theta)$ on watermarked text.
2. A forward and backward pass to compute $\nabla\mathcal{L}_{\text{anti}}(\pi_\theta)$ on anti-watermarked text.
3. A forward pass to evaluate $\mathcal{L}_{\text{anti}}(\pi_\theta - \alpha\hat{g})$, where $\hat{g}$ is the normalized forgetting gradient.

Thus, the overhead introduced by FPL is minimal. Moreover, CAWP accelerates convergence during training, making the overall computational cost comparable to baseline methods.

## C  COMPUTATIONAL EFFICIENCY ANALYSIS

To address concerns about computational efficiency, we provide a detailed comparison with prior watermarking methods. Our proposed techniques do not incur significant overhead and, in fact, accelerate convergence.

**CAWP.** The first component, CAWP, introduces only a lightweight trainable MLP as the watermark mapping model $M$, which contains 1.16M parameters—just $0.0388\%$ of a 3B-parameter LLM. This addition is negligible compared to the base LLM and does not substantially increase computational cost.

**FPL.** The second component, FPL, involves two additional forward and backward passes per iteration. However, by using a learning-friendly watermark, our method reduces the number of training steps needed to achieve high performance. For example, PRO reaches an AUC of 0.997 within 2000 steps, while KGW requires 5000 steps to reach a maximum AUC of 0.991.

**Wall-clock Training Time.** In practice, the total wall-clock training time remains comparable across methods. For 8B-parameter models trained on $4\times$A100 80GB GPUs, all methods (KGW, KTH, PRO) complete training in about 6 hours. Thus, despite the slightly higher per-iteration cost, PRO achieves better performance with no additional end-to-end training time.

## D  WATERMARKING FOR CLOSED-SOURCE LLMS

### D.1  KGW

The KGW watermark introduces a bias into the model's decoding distribution(Kirchenbauer et al., 2023). Given the model output distribution $p$, the watermarked distribution is defined as

$$f_w^{\text{KGW}} = \text{softmax}(\log p + \delta \cdot h(x_{t-k+1:t}; \xi, \gamma, |\mathcal{V}|)), \tag{52}$$

where $h(\cdot)$ produces a binary vector over the vocabulary that identifies the $\gamma$-portion of tokens to which the bias $\delta$ is applied. Its output is determined by a secret key $\xi$ and the previous $k$ tokens; when $k > 1$, an additive hash is applied, and when $k = 0$, a fixed green list is used(Zhao et al., 2024).

**Detection.** Given a sequence $x$, the detector reconstructs the green list that would have been active at each step and checks whether the produced token belongs to it:

$$S = \sum_{t=k+1}^{\text{len}(x)} h(x_{t-k:t-1})_{x_t}. \tag{53}$$

For normal (unwatermarked) text, the matched tokens should appear only at the rate implied by the green-list proportion $\gamma$. If $S$ is noticeably larger than this expected level, it indicates that the generation process systematically preferred green-listed tokens, indicating the presence of the watermark.

## D.2 KTH

The KTH watermark modifies decoding by selecting the token that best matches a key-dependent score. Given the model distribution $p$, the watermarked decoder is

$$f_w^{\text{KTH}}(p, x, \xi) = \text{onehot}\left(\arg\max_i \frac{\xi_i^{(\text{len}(x))}}{p_i}, |\mathcal{V}|\right), \tag{54}$$

where the key $\xi = (\xi^{(1)}, \ldots, \xi^{(m)})$ consists of $m$ vectors in $[0, 1]^{|\mathcal{V}|}$, each sampled independently. This construction deterministically chooses the token with the largest key-to-probability ratio.

To increase variability across generations from the same prompt, the key is shifted by an offset $\tau$ before decoding, producing

$$\xi' = (\xi^{(1+\tau \bmod m)}, \ldots, \xi^{(m+\tau \bmod m)}).$$

A parameter $s \in [1, m]$ specifies the number of allowed shifts, taken as evenly spaced values

$$\tau = \{\, i \cdot \lfloor m/s \rfloor \mid 0 \le i < s \,\}.$$

Larger $s$ broadens the range of possible outputs and increases diversity.

**Detection.** To evaluate a candidate sequence $x$, the detector computes the alignment cost

$$d(x, \xi) = \sum_{t=1}^{\text{len}(x)} \log\left(1 - \xi_{x_t}^{(t)}\right), \tag{55}$$

where lower values indicate that the text follows the key more closely and is therefore more likely watermarked. The significance of the observed score is assessed using a reference distribution obtained from non-watermarked text, and with a reference set of size $T$, the resulting $p$-value is bounded below by $1/(T + 1)$.

# E MODEL MODIFICATION

Open-source LLM are subject to modifications for task adaptation or deployment efficiency. The most prevalent types of such modifications include quantization, pruning, merging, and fine-tuning.

## E.1 QUANTIZATION

Model quantization is an important technique for running large language models on memory-constrained hardware. It reduces the precision of parameters and activations from formats such as FP32 or BF16 to integer representations like INT8 or INT4, helping to lower storage use and speed up inference.

Quantization approaches can be grouped into weight-only quantization and calibration-guided quantization. Weight-only methods apply fixed quantizers directly to model weights without data calibration,

such as LLM.int8() (Dettmers et al., 2022) and the NF4 format used in QLoRA (Dettmers et al., 2023). Calibration-guided methods optimize quantization parameters using a small calibration set to reduce reconstruction error. HQQ focuses on weight reconstruction, while GPTQ (Frantar et al., 2023) and AWQ (Lin et al., 2024) further reduce activation-level errors to improve the fidelity of the quantized model.

### E.2 PRUNING

Model pruning serves as a critical technique for optimizing large language models by systematically eliminating unnecessary parameters to achieve computational efficiency and reduced memory footprint. The fundamental principle involves identifying and removing weights that contribute minimally to model performance, thereby creating a more compact representation without substantial accuracy degradation.

Pruning strategies divide into two primary paradigms: unstructured and structured methodologies. In unstructured pruning, individual parameters are evaluated and removed based on their importance, creating irregular sparsity patterns throughout the network. This granular approach enables precise control over sparsity levels but often requires specialized hardware for efficient inference. Conversely, structured pruning eliminates coherent parameter groups—including entire channels, filters, attention mechanisms, or even complete layers—resulting in regular sparsity patterns that align naturally with existing hardware architectures.

The optimization objective in pruning centers on preserving the functional behavior of the original model $f(\cdot; \boldsymbol{\theta})$ after parameter removal. Given a calibration dataset $\mathcal{D}$, the pruning process seeks to identify sparse parameters $\boldsymbol{\theta}'$ that minimize the divergence between the original and compressed models:

$$\min_{\boldsymbol{\theta}'} \sum_{x \in \mathcal{D}} \|f(x; \boldsymbol{\theta}) - f(x; \boldsymbol{\theta}')\|^2, \tag{56}$$

Here, $\boldsymbol{\theta}'$ represents the parameter set after pruning, where numerous weights have been eliminated through zeroing (unstructured) or complete removal (structured).

Contemporary unstructured approaches like WANDA (Sun et al., 2024), SPARSEGPT (Frantar & Alistarh, 2023), and GBLM (Das et al., 2023) employ sophisticated scoring mechanisms to assess parameter significance at the individual weight level, enabling high sparsity ratios while maintaining model quality. Meanwhile, structured techniques including SHEARED LLAMA (Xia et al., 2024) and LLM-PRUNER (Ma et al., 2023) adopt a holistic approach, jointly optimizing the removal of architectural components to achieve efficient compression while adhering to the reconstruction objective.

### E.3 MODEL MERGING

Model merging represents an innovative paradigm for synthesizing multiple specialized models into a unified architecture by strategically combining their learned parameters. This approach enables the integration of diverse capabilities acquired through task-specific training, effectively consolidating expertise from different domains into a single deployable model without requiring additional training.

Recent investigations (Yang et al., 2024; Goddard et al., 2024) have revealed that parameter-space merging can successfully preserve and even amplify performance across multiple tasks, suggesting that distinct model capabilities can coexist within a shared parameter framework. The theoretical foundation rests on the notion of *task vectors*—the hypothesis that different model competencies manifest as approximately orthogonal directions within the high-dimensional parameter space, enabling their combination through various mathematical operations including weighted averaging and geometric interpolation.

Among the various merging strategies, *Spherical Linear Interpolation* (SLERP) emerges as a mathematically rigorous approach that respects the geometric structure of the parameter space. Unlike naive linear interpolation, SLERP traces the geodesic path between model parameters on the surface of a hypersphere, maintaining constant norm throughout the interpolation process. For two parameter configurations $\boldsymbol{\theta}_1$ and $\boldsymbol{\theta}_2$ separated by angle $\Omega$, the interpolated parameters at position

$t \in [0, 1]$ are computed as:

$$\text{SLERP}(\boldsymbol{\theta}_1, \boldsymbol{\theta}_2, t) = \frac{\sin[(1-t)\Omega]}{\sin \Omega} \boldsymbol{\theta}_1 + \frac{\sin[t\Omega]}{\sin \Omega} \boldsymbol{\theta}_2. \tag{57}$$

This spherical interpolation maintains the magnitude relationships inherent in the original models, potentially preserving learned representations more faithfully than alternative methods. When the parameter vectors align perfectly ($\Omega = 0$), the formulation gracefully degenerates to standard linear interpolation, ensuring computational stability across all configurations. The geometric preservation properties of SLERP make it particularly suitable for merging models where maintaining the relative structure of learned features is crucial for performance retention.

### E.4    FINE-TUNING

Model finetuning (Zhang et al., 2025; 2024) constitutes a fundamental adaptation methodology that transforms general-purpose language models into specialized systems through targeted training on domain-specific or task-oriented datasets. This technique bridges the gap between broad pretraining objectives and precise downstream requirements, enabling models to acquire specialized knowledge and behaviors absent from their initial training distribution.

The finetuning paradigm encompasses several distinct approaches, notably *Supervised Finetuning (SFT)* and *Instruction Tuning*. SFT directly optimizes model parameters using labeled examples from the target domain, establishing direct mappings between inputs and desired outputs. Instruction tuning extends this framework by training models to interpret and execute natural language directives across heterogeneous tasks, fostering more flexible and generalizable behavior. This instruction-following capability proves essential for developing conversational AI systems that can engage in open-ended dialogue while maintaining alignment with user intentions.

The optimization framework underlying both approaches seeks to minimize prediction error on the adaptation dataset. Starting from pretrained parameters $\boldsymbol{\theta}$, the finetuning process identifies updated parameters $\boldsymbol{\theta}'$ that optimize performance on the target dataset $\mathcal{D} = \{(x_i, y_i)\}_{i=1}^N$:

$$\min_{\boldsymbol{\theta}'} \frac{1}{N} \sum_{i=1}^N \mathcal{L}(f(x_i; \boldsymbol{\theta}'), y_i), \tag{58}$$

where $\mathcal{L}$ denotes the task-specific loss function measuring the discrepancy between model predictions and ground truth labels.

Contemporary finetuning strategies range from full-model updates to parameter-efficient alternatives. While comprehensive finetuning modifies all model parameters, resource-constrained scenarios benefit from selective adaptation techniques like LoRA (Low-Rank Adaptation) or adapter modules. These methods introduce minimal trainable parameters while freezing the majority of the pretrained model, achieving computational efficiency without compromising adaptation quality. Such parameter-efficient approaches have become increasingly vital as model scales continue to expand, enabling practical deployment of specialized models across diverse computational environments.

## F    EXPERIMENTS CONFIGURATION.

### F.1    BASELINE SETTING

Each table lists the hyperparameter configurations for the four open-source watermarking methods. The columns Left, Middle, and Right indicate the parameter settings used for the leftmost, middle, and rightmost points on each method's curve in the plots.

### F.2    TRAINING CONFIGURATIONS

All models are fine-tuned on subsets of OpenWebText with different watermark strategies using a batch size of 64 sequences, sequence length of 512 tokens, a maximum learning rate of 1e-5 with cosine decay, and a linear warmup over the first 10% of steps. We use the AdamW optimizer with $(\beta_1, \beta_2) = (0.9, 0.999)$ and no weight decay. For KTH watermark distillation, we follow the same

Table 5: Watermarking hyperparameter configurations for LLaMA-8B.

| Method | Left | Middle | Right |
|---|---|---|---|
| Gu et al. (KGW) | $n = 0, \delta = 1$ | $n = 1, \delta = 1$ | $n = 1, \delta = 2$ |
| Gu et al. (KTH) | $s = 4$ | $s = 2$ | $s = 1$ |
| Gloaguen et al. (KGW) | $n = 0, \delta = 1$ | $n = 1, \delta = 1$ | $n = 1, \delta = 2$ |
| **PRO** | $\delta = 0.3$ | $\delta = 0.5$ | $\delta = 1$ |

Table 6: Watermarking hyperparameter configurations for LLaMA-3.2-3B.

| Method | Left | Middle | Right |
|---|---|---|---|
| Gu et al. (KGW) | $n = 0, \delta = 1$ | $n = 1, \delta = 1$ | $n = 1, \delta = 2$ |
| Gu et al. (KTH) | $s = 4$ | $s = 2$ | $s = 1$ |
| Gloaguen et al. (KGW) | $n = 0, \delta = 1$ | $n = 1, \delta = 1$ | $n = 1, \delta = 2$ |
| **PRO** | $\delta = 0.5$ | $\delta = 1$ | $\delta = 1.5$ |

setup except for using a batch size of 128 and a sequence length of 256 tokens to accommodate memory constraints. For the Gloaguen et al. (Gloaguen et al., 2025)-KGW baseline, we follow their configuration. Starting from a model distilled with KGW, we fine-tune it on OpenWebText for 2,500 steps with a batch size of 64, sequence length of 512 tokens, a learning rate of 1e-5, and the AdamW optimizer. A cosine learning rate schedule is applied with 500 warmup steps. To identify stable parameters, we compute contrastive task vectors based on parameter change before and after fine-tuning, and perform a second-stage distillation restricted to these stable weights. Each training run took approximately 6 hours on 4 NVIDIA A100 80GB GPUs.

### F.3  MODEL MODIFICATION SETTINGS

To assess robustness, we apply four types of model modifications that simulate real-world user behavior: (1) quantization, including INT8 (Dettmers et al., 2022) and GPTQ (Frantar et al., 2023) at 8-bit precision, GPTQ(Frantar et al., 2023) and HQQ (Badri & Shaji, 2023) at 4-bit; (2) unstructured pruning using WANDA (Sun et al., 2024) and SparseGPT (Frantar & Alistarh, 2023) at 20% and 50% sparsity levels; (3) model merging via SLERP (Goddard et al., 2024), where the watermarked model is interpolated with its non-watermarked base model using mixing ratios from 0.1 to 0.9 following (Gloaguen et al., 2025); and (4) full-parameter fine-tuning on the task-specific OPENMATHINSTRUCT dataset (Toshniwal et al., 2024) and OPENCODEINSTRUCT dataset (Ahmad et al., 2025), reflecting the common use case where LLM users fine-tune open-source models on downstream data to build domain-specific experts. All modifications are implemented following the original settings of their respective methods.

## G  ADDITIONAL EXPERIMENTS

### G.1  MORE DATASET AND MODELS

We run evaluations on an additional large language model, GPT-J-6B, using the same settings as in Figure 6 of the main paper. The results in Figure 9 show that the proposed **PRO** consistently outperforms other methods in terms of watermark detectability (i.e., AUC) and generation quality (i.e., Perplexity). Specifically, to achieve an AUC above 0.99, **PRO** maintains a perplexity of 17.8, while existing methods require at least a perplexity of 21.5.

### G.2  VISUALIZATION OF ROC CURVES

In Figure 7, we visualize the ROC curves for the proposed **PRO** and the state-of-the-art method by Gu et al.. We select watermarked LLaMA3-8B models with the same level of perplexity (i.e., 4.7). The results indicate that **PRO** achieves better watermark detectability while maintaining the same level of generation quality.

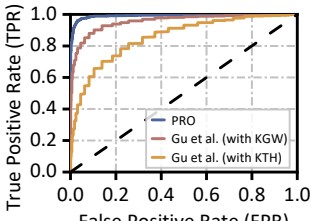

Figure 7: ROC curves for dif-

Table 7: Watermarking hyperparameter configurations for Phi-2-2.7B.

| Method | Left | Middle | Right |
|---|---|---|---|
| Gu et al. (KGW) | $n = 0, \delta = 1$ | $n = 1, \delta = 1$ | $n = 1, \delta = 2$ |
| Gu et al. (KTH) | $s = 4$ | $s = 2$ | $s = 1$ |
| Gloaguen et al. (KGW) | $n = 0, \delta = 1$ | $n = 1, \delta = 1$ | $n = 1, \delta = 2$ |
| **PRO** | $\delta = 0.5$ | $\delta = 1$ | $\delta = 1.5$ |

Specifically, the ROC curve of **PRO** closely follows the top-left corner, indicating a higher true positive rate (TPR) across nearly all false positive rate (FPR) thresholds. In contrast, the methods by Gu et al. show noticeably lower TPRs, particularly at low FPR regions, which suggests a less reliable distinction between watermarked and non-watermarked texts under tight detection constraints. This improvement is especially evident in the early phase of the curve (e.g., FPR $< 0.1$), where **PRO** already achieves near-perfect detection performance, while the baseline methods lag behind. These findings further confirm that **PRO** not only embeds robust watermark patterns but also enables more confident detection, even at low error tolerance levels, making it more suitable for security-critical applications.

### G.3 OVERALL DETECTION AUC

We compared our **PRO** and baselines in terms of detection AUC and perplexity in Figure 8, the results shown that our method **PRO** can achieve the best AUC while maintaining the highest perplexity.

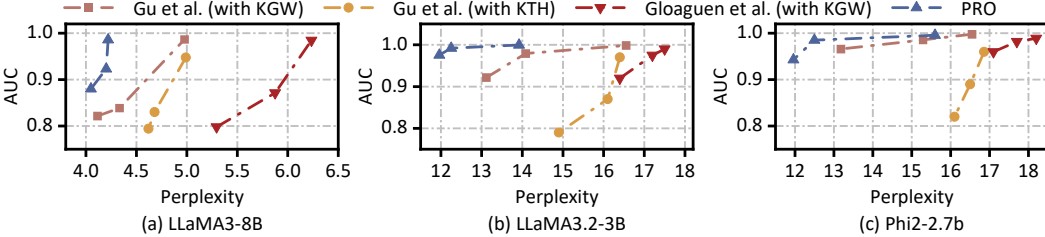

Figure 8: Effectiveness comparison of different open-source LLMs text watermarks, in terms of detection AUC and median PPL on three LLMs. A better watermark detectability and generation quality is indicated by higher AUC and lower PPL, as shown by lines closer to the upper left corner.

### G.4 DETECTION PERFORMANCE UNDER DIFFERENT FPRS

We further evaluate detection robustness by measuring the true positive rate (TPR) at different false positive rate (FPR) levels. This complements the AUC metric and highlights performance in stricter detection regimes. As shown in Table 8, **PRO** consistently outperforms prior methods, especially under very low FPRs (e.g., 0.1%), which are critical for practical deployment.

Table 8: Comparison of detection performance (TPR at different FPR levels) on LLaMA-3-8B. **PRO** demonstrates significantly stronger detection under stringent low-FPR conditions.

| Method | TPR@0.1% ↑ | TPR@1% ↑ | TPR@10% ↑ |
|---|---|---|---|
| Gu et al. (KTH) | 25.8% | 53.6% | 84.0% |
| Gu et al. (KGW) | 54.2% | 82.7% | 97.5% |
| **PRO** | **78.1%** | **92.3%** | **99.5%** |

We run evaluations on an additional dataset of Wikipedia articles, using the same settings as in Figure 6 of the main paper, except for the dataset. We evaluate 5,000 completions of 200 tokens each,

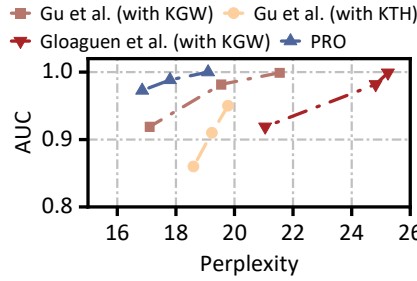

Figure 9: Results on GPT-J-6B.

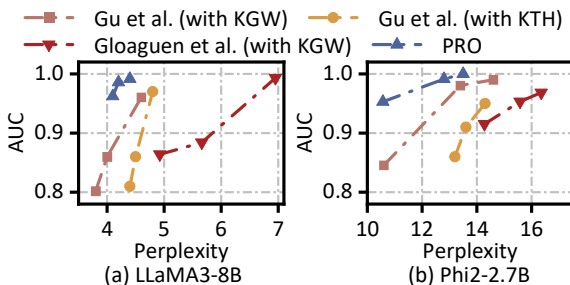

Figure 10: Results on Wikipedia dataset.

generated from 50-token prompts. As shown in Figure 10, the proposed **PRO** consistently achieves better watermark detectability at the same level of generation quality, indicating its generalizability to downstream users' prompts.

### G.5 ROBUSTNESS AGAINST MODEL MODIFICATIONS ACROSS DOWNSTREAM TASKS

We conducted experiments to evaluate FPL's effectiveness on more diverse downstream tasks. Specifically, we fine-tuned on the Alpaca dataset to further examine watermark robustness beyond code and math domains. As shown in Table 9, our method consistently outperforms the best baseline (Gloaguen et al. with KGW), with PRO's relative improvements shown in parentheses.

Table 9: Performance of watermarked LLaMA-3-8B under different fine-tuning steps $s$. The values in parentheses (prefixed with $+$) indicate the relative improvement compared to the baseline.

| Step $s$ | AUC | TPR@1% | TPR@10% |
|---|---|---|---|
| 300 | 0.91 (+0.02) | 54.3% (+2.8%) | 72.5% (+6.0%) |
| 600 | 0.87 (+0.04) | 49.4% (+4.0%) | 69.9% (+2.9%) |
| 900 | 0.85 (+0.02) | 40.5% (+9.1%) | 60.2% (+6.9%) |
| 1200 | 0.81 (+0.04) | 26.8% (+13.0%) | 44.7% (+13.6%) |
| 1500 | 0.79 (+0.05) | 16.9% (+16.5%) | 33.3% (+21.0%) |

### G.6 COMPARED WITH CLASSIFIER-BASED LLM TEXT DETECTORS

One distinct advantage of watermarking over classifier-based detectors is its ability to attribute text to a specific model, rather than just distinguishing LLM and human text. We evaluated PRO using the open-source classifier SuperAnnotate/ai-detector, which flagged 90.6% of PRO-generated texts as AI-generated. It also flagged 90% of outputs from non-watermarked and differently watermarked LLMs(Qwen-4B/8B, GPT-6B, LLaMA3-3B) as LLM-generated. To further analyze this, we fine-tuned a binary RoBERTa classifier on texts from LLaMA3-8B and human texts. While it detected 60–80% of outputs from other LLMs (Qwen-4B/8B, GPT-6B, LLaMA3-3B) as LLaMA3-8B-generated. This shows that classifier-based methods primarily learn to distinguish LLM vs. human text, not between different LLMs In contrast, our PRO watermark enables model-level attribution. Specifically, we used the watermark policy model trained with LLaMA3-8B (student LLM) to detect outputs from other LLMs. The false positive rates remained low: 0.8% (Qwen-4B), 1.0% (Qwen-8B), 1.1% (GPT-6B), and 0.6% (LLaMA3-3B). This highlights PRO's precision in determining whether a text was generated by a specific model.

### G.7 DIFFERENT EMBEDDING MODEL AND WATERMARK MAPPING MODEL

To further validate the generality of our method, we supplemented the experiments with two additional embedding models: `thenlper/gte-large` and `intfloat/e5-large-v2`. Results in Table 10 show that our method maintains strong detection performance across different embeddings, consistently achieving near-perfect AUC and high TPR scores.

Table 10: Performance of PRO watermarking with LLaMA-3.2-3B under different embedding models.

| Embedding Model | AUC | TPR@1% | TPR@10% |
|---|---|---|---|
| thenlper/gte-large | 0.997 | 95.2% | 99.2% |
| intfloat/e5-large-v2 | 0.994 | 94.7% | 99.0% |

We also compared different architectures for the watermark mapping model $M$. An MLP yields the best performance, as it can effectively exploit the full semantic embedding. By contrast, convolutional neural networks (CNNs) perform significantly worse, since semantic embeddings are single dense vectors without spatial or sequential structure for convolution to leverage. Thus, we adopt the MLP design in our framework.

### G.8 CONFIGURABLE WATERMARK TOKEN RATIO $\gamma$

In the main text, the watermark mapping model $M$ is trained to produce logits that are approximately centered around zero, which results in an effective green token ratio close to $0.5$. This choice is made for simplicity and does not limit the flexibility of the framework. PRO also supports arbitrary watermark token ratios $\gamma \in (0, 1)$.

To target a desired ratio $\gamma$, we apply a simple scaling to the raw watermark logits. Let $v_i = M(e_i) \in [-1, 1]^{|\mathcal{V}|}$ denote the watermark logits corresponding to the embedding $e_i$. We define a transformed logit vector $s_i = [s_{i,j}]_{j=1}^{|\mathcal{V}|}$ as:

$$s_{i,j} = \begin{cases} \frac{1-\gamma}{\gamma}\, v_{i,j}, & v_{i,j} > 0, \\ v_{i,j}, & \text{otherwise.} \end{cases}$$

During training, the normalization objective is applied to $s_i$ in place of $M(e_i)$, encouraging the scaled logits to realize the specified green token proportion $\gamma$. The similarity objective $\mathcal{L}_{\text{sim}}$ remains unchanged.

At detection time, the same scaling is applied before computing the watermark score, and the expected value in the $z$-test is updated from $0.5$ to $\gamma$. No other modifications to the detection procedure are required.

