# OpenReview forum: "PRO: Enabling Precise and Robust Text Watermark for Open-Source LLMs"
_ICLR.cc/2026/Conference — Submitted to ICLR 2026_

### Official Review · Reviewer_2HJA · 2025-10-31

**Soundness:** 2
**Presentation:** 3
**Contribution:** 2
**Rating:** 4
**Confidence:** 4

**Summary:**

This paper proposes PRO, a framework for embedding text watermarks directly into the weights of open-source large language models.
It introduces two key components: 1) Co-Adaptive Watermark Policy (CAWP), which jointly trains a learnable watermark mapping and the model to improve generation–detection consistency, and 2) Forgotten Perturbation-aware Learning (FPL), which simulates fine-tuning perturbations during training to enhance robustness against model modifications such as merging, pruning, and quantization.
Experiments on LLaMA and Phi models show that PRO achieves high AUC and low perplexity while maintaining strong robustness compared to prior watermarking methods.

**Strengths:**

The paper proposes a watermarking for open-source LLMs. The proposed PRO framework—with its co-adaptive watermark policy (CAWP) and forgotten perturbation-aware learning (FPL) is empirically effective, showing strong robustness against fine-tuning, merging, and pruning while maintaining good text quality.

**Weaknesses:**

1. The convergence analysis in Appendix D essentially restates the standard smoothness and gradient-descent convergence results of KL-based distillation. However, it does not establish any theoretical link between the proposed objectives and the final detection metrics (e.g., AUC or TPR at low FPR), nor does it formally connect to the claimed robustness objective (detectability under fine-tuning, merging, or pruning). For FPL, what is the bound the forgotten perturbation space? In reality, users perform diverse modifications such as LoRA/adapters, parameter averaging or smoothing, and even full re-distillation. It remains unclear how this simplified perturbation approximates such real-world updates.

2. In Table 1, the paper reports AUC which measures overall separability across all thresholds and TPR @ 5%FPR. However, in watermark detection, the low-FPR region is the only practically relevant regime but high AUC does not imply good performance at low FPR. Table 1 should report TPR @ 1% FPR (and ideally TPR @ 0.1\% FPR). This would align with common practice in watermarking papers (e.g., KGW [1]) and make the results interpretable for real-world deployment.

3. The robustness evaluation currently covers only full-parameter fine-tuning, model merging, quantization, and pruning.
However, in the open-source ecosystem, attackers typically use cheaper and more practical operations such as distillation, i.e., training a new student model using an nonwatermarked teacher. This strong modification could completely rewrite the output distribution and is feasible for professional attackers.

[1] A Watermark for Large Language Models. John Kirchenbauer, Jonas Geiping, Yuxin Wen, Jonathan Katz, Ian Miers, Tom Goldstein. ICML 2023

**Questions:**

See Weaknesses

---

> ### Author Response · Authors · 2025-11-25
> **Response to Reivewer 2HJA's concerns (1/2)**
>
> We thank Reviewer 2HJA for the detailed and constructive feedback. We provide the following response to the reviewer’s question:
>
> > **Q1. Formal analysis of improvements of CAWP and FPL.**
>
> Developing a precise and robust watermarking framework for open-source LLMs is new and critical to the watermark community, and our results and benchmarks provide a practical solution. To further ensure the method is not only practically effective but also theoretically grounded, we provide a formal analysis as follows. The improvement stems from minimizing the similarity loss $\mathcal{L}\_{sim}$ in an expanded optimization space:
>
> - Prior Work: fix the watermark teacher, optimizing only the student LLM $\theta$: $\min\_{\theta} \mathcal{L}\_{sim}(\pi\_{\theta},\pi\_{fixed})$. This limits the optimization to fitting a static, predefined target.
>
> - Ours: CAWP employs a trainable watermark policy $M_{\phi}$, formulating a joint optimization: $\min\_{\theta, \phi} \mathcal{L}\_{sim}(\pi\_{\theta}, \pi\_{\phi})$. By expanding the optimization space to include $\phi$, we allow the system to discover watermark patterns that are inherently easier for the LLM to learn. Consequently, we achieve a lower minimum for $\mathcal{L}_{sim}$ (KL divergence) compared to the fixed setting.
>
> A lower $\mathcal{L}\_{sim}$ indicates a tighter alignment between the student and the watermark distribution, leading to higher detectability. Please refer to Appendix B.2 for the detailed convergence analysis.
>
> Similarly, for robustness (FPL), the improvement comes from explicitly regularizing the model's sensitivity to weight drift:
>
> - Standard Training: Conventional methods optimize $\pi_{\theta}$ solely for the watermark detectability, ignoring how future weight modifications might degrade it.
> - Ours (Perturbation-Aware): FPL introduces a regularization term (in Eq.6) that simulates a perturbation step along the forgetting direction. By minimizing the loss degradation caused by this simulated perturbation during training, we force the model to converge to a "flat region" where the watermark remains stable even after significant weight shifts.
>
> For the detailed derivation of this second-order approximation and its computational efficiency, please refer to Appendix B.3.
>
> > **Q2. What is the bound the forgotten perturbation space? It remains unclear how this forgotten perturbation approximates such real-world updates.**
>
> The "bound" of the forgotten perturbation space is defined theoretically and controlled empirically:
>
> - Theoretical Bound: In our theoretical analysis (Appendix B.3), we define the bound as a radius $R$ in the parameter space, such that any user modification $\Delta$ satisfies $||\Delta||\le R$. Our goal is to minimize the worst-case drop in watermark detectability within this radius.
> - Empirical Control ($\alpha$ & $\beta$): In Eq. 6, we explicitly control this bound via the perturbation step size $\alpha$, which defines the radius of the simulated weight drift, while the regularization weight $\beta$ controls the strength of the model's resistance within that space. As shown in Table 4, our ablation study confirms that introducing this perturbation mechanism with an appropriate $\alpha$ is essential for enhancing robustness against model modifications compared to standard training.
> - Approximation of Real-World Updates: Our perturbation $\hat{g} = \nabla \mathcal{L}_{anti}$ targets the worst-case scenario rather than specific user modifications. As shown in Section 3.3, gradients from anti-watermarked texts represent the "direction of maximum destruction," causing watermark collapse. By training the model to resist this dominant threat, we effectively enhance its robustness against diverse downstream modifications [a]. This approach is backed by our theoretical analysis (Section B.3) and validated empirically

---

> ### Author Response · Authors · 2025-11-25
> **Response to Reivewer 2HJA's concerns (2/2)**
>
> > **Q3. Table 1 should report TPR@1% FPR (and ideally TPR @0.1% FPR).**
>
> In the revised manuscript, we have replaced AUC with TPR@1FPR in Table 1 and Figure 6, and with TPR@0.1FPR in Table 6. The results show that PRO’s detectability and robustness advantages remain consistent under these stricter metrics.
>
> > **Q4. Watermark roboustness against the distillation attack.**
>
> Evaluating watermark robustness comprehensively remains an open challenge in the field. Prior work has primarily examined robustness under common model modifications such as quantization, pruning, model merging, and fine-tuning. Following this standard practice, we show that PRO achieves enhanced robustness across all of these settings. This is because PRO’s improvements arise from general principles rather than task-specific defenses: FPL explicitly optimizes against perturbations that most weaken watermark signals, and CAWP produces learning-friendly patterns that persist more effectively through model adaptation, including distillation.
>
> Per the reviewer’s suggestion, we added new experiments evaluating distillation-based watermark removal. Specifically, we distilled a smaller LLaMA3-3B model using text outputs from a larger LLaMA3-8B model watermarked with either PRO or Gu et al.–KGW. Using 1.6M watermarked tokens and identical training settings for both methods (batch size 32, block size 256, 200 steps), we observe that the student distilled from PRO-watermarked data inherits the watermark and remains detectable, whereas the student distilled from KGW-watermarked data shows noticeably weaker transfer. These results indicate that PRO causes distillation to more easily inherit the watermark signal, thereby frustrating attackers attempting to remove it through distillation and further complementing its robustness against other model modifications.
>
> TPR@1%FPR under distillation
> | Method     |  Step 40   |  Step 80  |  Step 120 |   Step 160   |   Step 200     |
> |-----------------------|------|-----------|---------------|------------------|-------|
> | Gu et al. (2023)-KGW| 0.01  | 0.02    | 0.05     |  0.07     |0.11  |
> | PRO| 0.05  |  0.21    |  0.51     |  0.61    |0.71  |
>
>
>
> [a]  Madry, Aleksander, et al. "Towards Deep Learning Models Resistant to Adversarial Attacks." ICLR 2018.

---

### Official Review · Reviewer_LnXy · 2025-10-31

**Soundness:** 3
**Presentation:** 3
**Contribution:** 3
**Rating:** 6
**Confidence:** 3

**Summary:**

This paper introduces PRO, a Precise and Robust watermarking framework for open-source large language models (LLMs) that embeds watermarks directly into model weights rather than at the decoding stage, making it effective even when users modify or re-implement decoding. The approach features a trainable watermark policy model jointly optimized with the base LLM to encourage watermark-friendly generation patterns, and a robustness regularization term that simulates downstream perturbations such as fine-tuning and model merging while penalizing any degradation in watermark detectability. Experiments on LLaMA-3.2, LLaMA-3, and Phi-2 show that PRO achieves higher watermark detectability and stronger robustness than prior methods, with publicly released code supporting reproducibility.

**Strengths:**

- The paper addresses a practical and growing problem: watermarking open-source LLMs where owners lack control over decoding.
- The experiments across multiple open-source models demonstrate that PRO yields higher watermark detectability and improved resistance to post-training modification compared to baseline methods.
- The paper is clearly structured, with intuitive figures.

**Weaknesses:**

- The paper does not provide a formal analysis or theoretical guarantee on why the joint optimization leads to higher detectability or robustness. A more rigorous treatment (e.g., gradient alignment or mutual information perspective) would strengthen the claims.
- While PRO aims for “precise and robust” watermarking, the authors do not systematically evaluate how the approach affects the model’s general performance (e.g., perplexity, generation quality, reasoning accuracy) on more diverse and broader tasks.
- The paper does not isolate the contributions of the trainable policy model and the robustness regularizer.
- While PRO simulates perturbations from fine-tuning and model merging, it overlooks other crucial real-world perturbations, such as reinforcement learning–based post-training (e.g., RLHF or DPO).

**Questions:**

In general, the paper is very comprehensive. Please refer to the weakness part.

---

> ### Author Response · Authors · 2025-11-25
> **Response to Reivewer LnXy's concerns**
>
> We thank Reviewer LnXy for the positive review and feedback.
>
> > **Q1. Formal analysis of improvements of CAWP and FPL.**
>
> Developing a precise and robust watermarking framework for open-source LLMs is new and critical to the watermark community, and our results and benchmarks provide a practical solution. To further ensure the method is not only practically effective but also theoretically grounded, we provide a formal analysis as follows. The improvement stems from minimizing the similarity loss $\mathcal{L}\_{sim}$ in an expanded optimization space:
>
> - Prior Work: fix the watermark teacher, optimizing only the student LLM $\theta$: $\min\_{\theta} \mathcal{L}\_{sim}(\pi\_{\theta},\pi\_{fixed})$. This limits the optimization to fitting a static, predefined target.
>
> - Ours: CAWP employs a trainable watermark policy $M_{\phi}$, formulating a joint optimization: $\min\_{\theta, \phi} \mathcal{L}\_{sim}(\pi\_{\theta}, \pi\_{\phi})$. By expanding the optimization space to include $\phi$, we allow the system to discover watermark patterns that are inherently easier for the LLM to learn. Consequently, we achieve a lower minimum for $\mathcal{L}_{sim}$ (KL divergence) compared to the fixed setting.
>
> A lower $\mathcal{L}\_{sim}$ indicates a tighter alignment between the student and the watermark distribution, leading to higher detectability. Please refer to Appendix B.2 for the detailed convergence analysis.
>
> Similarly, for robustness (FPL), the improvement comes from explicitly regularizing the model's sensitivity to weight drift:
>
> - Standard Training: Conventional methods optimize $\pi_{\theta}$ solely for the watermark detectability, ignoring how future weight modifications might degrade it.
> - Ours (Perturbation-Aware): FPL introduces a regularization term (in Eq.6) that simulates a perturbation step along the forgetting direction. By minimizing the loss degradation caused by this simulated perturbation during training, we force the model to converge to a "flat region" where the watermark remains stable even after significant weight shifts.
>
> For the detailed derivation of this second-order approximation and its computational efficiency, please refer to Appendix B.3.
>
> > **Q2. How the approach affects the model’s general performance (e.g., generation quality) on more diverse and broader tasks?**
>
> We thank the reviewer for the suggestion to evaluate overall model utility. Following the comment, we additionally evaluated general downstream performance using lm-eval-harness across representative understanding and reasoning tasks. As shown below, PRO introduces negligible degradation (<0.3%) across all benchmarks, which is on par with or smaller than the variation introduced by baseline learning-based watermarking (Gu et al.-KGW).
>
> | Model                       | HellaSwag | ARC-Challenge | AGIEval (MathQA) | GSM8K |
> |-----------------------------|-----------|---------------|------------------|-------|
> | Base (Llama-3-8B-Instruct)  | 0.7607    | 0.5708        | 0.2963           | 0.7583|
> | Gu.et.al-KGW                | 0.7583    | 0.5683        | 0.2877           | 0.7582|
> | PRO (Ours)                  |  0.7594   | 0.5705        | 0.2920           | 0.7582|
>
>
> > **Q3. The paper does not isolate the contributions of the trainable policy model and the robustness regularizer.**
>
> In our ablation study, the **α = 0** setting already shows that **CAWP alone can significantly improve watermark robustness**, and enabling **β > 0** further enhances robustness by introducing FPL. Since FPL relies on CAWP to boost robustness, it cannot function meaningfully as a standalone component, so we did not include a separate “FPL-only” experiment. We will clarify this point more explicitly in the revised version.
>
> > **Q4. While PRO simulates perturbations from fine-tuning and model merging, it overlooks other crucial real-world perturbations, such as reinforcement learning–based post-training (e.g., RLHF or DPO).**
>
> To address this, we conducted additional experiments where the watermarked model is post-trained using a standard DPO-based RLHF setup on the HH-RLHF dataset. We trained with a batch size of 64 and a learning rate of 1e-6. In our observations, RLHF introduces smaller parameter perturbations than full fine-tuning on domain-specific dataset, and PRO maintains stable robustness for 5000 steps. We report TPR@5FPR in the following table. The baseline (Gu et al.-KGW) exhibits a slight drop under the same setting. We thank the reviewer for highlighting this scenario and will include a discussion on RLHF post-training in the revised version.
>
> | Model                | Step=1000  | Step=2000      | Step=3000         | Step=4000 |Step=5000 |
> |-----------------------|-----------|---------------|------------------|-------|----------------|
> | Gu.et.al-KGW         |   0.954      |   0.949       |  0.938    |  0.927     | 0.922  |
> | PRO (Ours)            |  0.983        |   0.981       | 0.977     |   0.971    | 0.964  |

---

### Official Review · Reviewer_gnTk · 2025-10-31

**Soundness:** 3
**Presentation:** 3
**Contribution:** 3
**Rating:** 6
**Confidence:** 3

**Summary:**

This paper proposes a framework to embed resilient watermarks directly into model weights. It introduces Co-Adaptive Watermark Policy (CAWP) to jointly learn watermark patterns aligned with the model’s behavior, reducing generation–detection inconsistency, and Forgotten Perturbation-aware Learning (FPL) to enhance robustness against fine-tuning and model merging. Experiments on LLaMA3 and Phi2 models show that PRO achieves higher detectability, better text quality, and stronger robustness than existing open-source watermarking methods.

**Strengths:**

1. Identify the problem of Generation-Detection Inconsistency. The mappings of watermarked tokens are arbitrary.
2. Provide a novel method co-adapting the watmeark model with the real model to better align the watermark with the model's innate performance. And innovatively devise the FPL module to properly solve the weakness of the current open-source model watermark to finetuning.
3. carry out experiment validating the performance of PRO.

**Weaknesses:**

1. Using model merging as an attack to evaluate learning-based watermarking may be inappropriate, since such attacks assume access to an unwatermarked model. In my opinion model merging shouldn't be considered as a valid attack.
2. Because a key component of CAWP relies on an MLP that extracts semantic information through a BERT encoder, it would be important to include comparisons with prior semantic-invariant distillation method to demonstrate the necessity and contribution of the co-training design.
3. The FPL's loss function seems to only prevent local curvature, lack theoretical and empiraicla exepriment for its efficacy when the model's being trained on more anti-watemrakr token. Does it means the watermakred model itself is hard to be finetuned and gain downstream capability?

**Questions:**

1. The experiment detail is missing in 3.2's motivation study, which shows the inconsistency between the teacher model and student model, According to prior work [1], it seems to me that the reported AUC drop is unexpectedly large and may require further clarification or replication details.

2. While AUC is reported as the main detection metric, it is not the most interpretable measure for practical watermark detection. It is recommended to include TPR at a given FPR in the main paper rather than relegating it to the appendix, as this provides a clearer assessment of real-world detection performance.
(also see my questions in weakness)

[1] Gu, C., Li, X. L., Liang, P., & Hashimoto, T. (2024). On the Learnability of Watermarks for Language Models. In Proceedings of the Twelfth International Conference on Learning Representations (ICLR 2024).

---

> ### Author Response · Authors · 2025-11-25
> **Response to Reivewer gnTk's concerns**
>
> We thank Reviewer gnTk for the positive review and feedback.
>
> > **Q1. Using model merging as an attack to evaluate the robustness of learning-based watermarking may be inappropriate.**
>
> Model merging robustness is actually a strength of PRO, and prior work has also evaluated this scenario (e.g., Gloaguen et al. 2025, Table 1). We agree that model merging is not a malicious attack, but rather a legitimate and common practice in the open-source LLM ecosystem, making watermark robustness to merging essential for practical deployment. Users routinely merge specialized models (e.g., LLaMA3-Math) using tools like MergeKit to combine capabilities, and may unknowingly merge watermarked with unwatermarked models. LLM developers need the watermark to remain detectable after downstream merges, enabling them to verify model provenance and protect their intellectual property despite legitimate user-driven modifications. That's why both of prior work and PRO study the watermark robustness against model merging.
>
> > **Q2. Comparisons with prior semantic-invariant distillation method to demonstrate the contribution of the co-training design.**
>
> Our CAWP addresses the generation-detection inconsistency problem in open-source LLMs where the student model fails to accurately learn the teacher's watermark patterns during distillation. Unlike prior methods that use semantic-invariant watermarking to combat paraphrasing attacks in closed-source settings, our approach **jointly** trains the MLP with the student LLM to generate watermark patterns that are inherently learnable, even though our MLP also processes semantic embeddings.
>
> To isolate the effect of co-optimization, we compare CAWP with the semantic-invariant SIR method under the same open-source distillation setup. Although SIR modestly improves detectability, a clear generation–detection mismatch remains (TPR@1%FPR improves only from 0.83 to 0.86, TPR@0.1%FPR improves from 0.54 to 0.64). In contrast, CAWP’s joint optimization of the watermark mapping model and the student LLM yields a larger gain, raising TPR@1%FPR from 0.86 to 0.92, TPR@0.1%FPR improves from 0.64 to 0.78. This shows that resolving the inconsistency comes from co-optimization, not from semantic embeddings alone. Moreover, CAWP better preserves model quality (PPL 4.2 vs. 4.6 for [a]).
>
> > **Q3. The FPL's loss function seems to only prevent local curvature. Does it means the watermakred model itself is hard to be finetuned and gain downstream capability?**
>
> FPL is not intended to prevent fine-tuning or suppress the model’s ability to learn new tasks. Its objective only reduces sensitivity **along the specific forgetting direction**—the gradient that most strongly pushes the model toward the *anti-watermark* distribution. In other words, FPL flattens curvature *only in the subspace that erases the watermark*, not in the full parameter space required for downstream learning.
>
> Empirically, we verify this in our robustness experiments (Table 1): even after plenty of fine-tuning steps on OpenMathInstruct and OpenCodeInstruct, the watermarked models continue to improve task performance (lower PPL), while retaining significantly higher AUC than baselines—indicating that downstream adaptation remains intact.
>
> Theoretically (Appendix B.3), the FPL bound shows that the regularizer only constrains perturbations within a bounded radius $R$ and only reduces the gradient norm associated with watermark forgetting, rather than inhibiting arbitrary optimization directions. Thus, FPL does not make the model “hard to fine-tune”; it simply makes it harder to fine-tune *in a way that unintentionally deletes the watermark*.
>
> > **Q4. The experiment detail is missing in 3.2's motivation study, which shows the inconsistency between the teacher model and student model. According to prior work, it seems to me that the reported AUC drop is unexpectedly large and may require further clarification or replication details.**
>
> The AUC drop of baseline is expected given our moderate watermark strength $(n=1, \delta=1)$ to preserve generation quality, it could achieve higher detectability with $\delta=2$ like [2], but this degrades quality significantly (PPL increases from 4.1 to 5.0 as shown in our Figure 2). Additionally, LLaMA3-8B's $4\times$ larger vocabulary (128K) compared to LLaMA2-7B (32K) used in [2] makes watermark learning in our experimental setting inherently more challenging.
>
> > **Q5. It is recommended to include TPR at a given FPR in the main paper as this provides a clearer assessment of real-world detection performance.**
>
> In the revised manuscript, we replace AUC with TPR@1FPR in Table 1 and Figure 6, and with TPR@0.1FPR in Table 6. The results show that PRO’s detectability and robustness advantages remain consistent under these stricter metrics.
>
> [a] A Semantic Invariant Robust Watermark for Large Language Models. ICLR 2023
>
> [b] On the Learnability of Watermarks for Language Models. ICLR 2023

---

### Official Review · Reviewer_2PZW · 2025-10-31

**Soundness:** 2
**Presentation:** 2
**Contribution:** 3
**Rating:** 4
**Confidence:** 4

**Summary:**

This paper develops a text watermarking method for open-weight LLMs. Because users have control over inference procedures, a watermark for open-weight LLMs must be learned into the weights, instead of relying on decoding-based watermarking. Towards this goal, this paper proposes PRO, a method which jointly trains a watermark policy alongside the LLM, and also includes a loss term that penalizes a decrease in watermark detectability under a weight perturbation. Experiments claim that PRO outperforms baseline methods in terms of text quality, watermark detectability, and robustness against model modifications (e.g., fine-tuning, model merging).

**Strengths:**

1. Effective and robust watermarking for open-weight LLMs is an important open problem. As open-weight LLMs become more capable and widely used, combating LLM misuse via methods such as watermarking become more important.
2. The proposed method seems like a natural way to approach the problem. It simultaneously optimizes the watermark policy to increase detectability, along with optimizing against degradation in detectability from a simulated gradient update step on red tokens.
3. The code is publicly released, enhancing the transparency and reproducibility of the results. Training configurations and hyperparameters are also included in the appendix.

**Weaknesses:**

1. Watermark detectability still drops significantly in PRO after fine-tuning. The TPR@5 decreases from 0.99 to 0.37 after 1500 fine-tuning steps on OpenMath Instruct, which I’m not sure I would call “robust”.
2. The numbers reported for the Gloaguen et al. (2025) method in Table 1 do not match up with the numbers they reported, even though the experimental setups seem to be mostly the same. [Gloaguen et al. (2025)](https://arxiv.org/abs/2502.10525) (Table 1\) reports 0.69 TPR@5 after 2,500 fine-tuning steps on OpenMathInstruct (and considers this **not** robust), whereas the PRO paper reports 0.222 TPR@5 after 1,500 fine-tuning steps on OpenMathInstruct. This significant discrepancy is strange.
3. [Gloaguen et al. (2025)](https://arxiv.org/abs/2502.10525) find that watermark durability can be improved by increasing the distillation dataset size. In their Table 2, they report that this method can achieve 0.91 TPR@5 after 2,500 fine-tuning steps on OpenMathInstruct. The PRO paper does not mention this method at all, and does not run any experiments on how the dataset size/number of training tokens affects durability.
4. The baseline perplexity of the original model before training is not included. It would be good to have the original model in Table 1, Figure 6, etc. in order to evaluate how much PRO impacts the text quality, compared to no watermarking. It might also be nice to compare with the performance of decoding-based watermarking (perhaps slightly less necessary).
   * Line 144 claims that PRO “can even match the performance of closed-source counterpart.” But I do not see any experimental comparisons with decoding-based watermarking to support this claim.
5. Watermark detection is now more expensive, as it requires running an embedding model and MLP. Most existing detectors are non-neural and can be run on CPU only.
6. It would be ideal to have I.4 Robustness Against Paraphrasing Attack in the main paper, as robustness to text modifications is an important aspect of evaluating watermarking methods. But I understand that the main paper is already at the page limit.
7. Appendix F: "Watermarking for Closed-Source LLMs" in this paper appears to be mostly taken from Appendix A: "Additional Details on Watermarking Strategies" in [Gu et al. (2023)](https://arxiv.org/abs/2312.04469), yet there is no citation. Some parts are nearly verbatim identical. A citation appropriately indicating the source and extent of reuse should be added to avoid potential plagiarism concerns.
   * Similarly, some parts of Appendix G: "Model Modification" appear to be paraphrased from Section 3: "Durability of Open-Source LLM Watermarks" of [Gloaguen et al. (2025)](https://arxiv.org/abs/2502.10525), again with no citation. The overall structure/organization of topics is very similar. An appropriate citation should also be added here.

**Questions:**

1. Why doesn’t the PRO method appear in Figure 2 (right)?
2. There should be citations for the models used, e.g., Llama 3 (line 139), Phi2 (line 140), BERT (line 264), etc.
3. Equation 3: KL divergence should be computed on the probabilities, not the raw logits. Also, $\\pi$ represented probabilities earlier, so it is inconsistent to have it denote logits now.
4. Line 280 describes ensuring half the tokens are more likely and the other half are less likely, i.e., $\\gamma \= 0.5$. But what if the developer wants to use some other value of $\\gamma$, such as 0.25?
   * Also, zero mean does not necessarily ensure that half are positive and half are negative. If $1/3$ are positive with logits 1 and $2/3$ are negative with logits \-0.5, then the mean is still zero.
5. The labels in equation (4) for the terms for (i) unbiased token preference and (ii) balanced watermark logits appear to be swapped.
   * The first term incentivizes zero mean across the vocabulary for each input embedding. (label should be balanced watermark logits)
   * The second term incentivizes each token to have zero mean across input positions. (label should be unbiased token preference)
6. Another related work that uses neural networks to generate watermark logits is [Liu et al. (ICLR 2024\)](https://openreview.net/forum?id=gMLQwKDY3N).
7. What is the perplexity of the original model before any training? And the original model with decoding-based watermarking?
8. How many tokens are the methods in Table 1 trained on?
9. What are the training details for the OpenMath/OpenCode fine-tuning experiments, e.g., batch size, learning rate, training time, etc.?
10. Several references are cited as arXiv preprints, but they have been published in conferences. For example, [Frantar el at. (ICLR 2023\)](https://openreview.net/forum?id=tcbBPnfwxS), [Gu et al. (ICLR 2024\)](https://openreview.net/forum?id=9k0krNzvlV), [Kirchenbauer et al. (ICLR 2024\)](https://openreview.net/forum?id=DEJIDCmWOz), [Kuditipudi et al. (TMLR)](https://openreview.net/forum?id=FpaCL1MO2C), [Sun et al. (ICLR 2024\)](https://openreview.net/forum?id=PxoFut3dWW), [Zhao et al. (ICLR 2024\)](https://openreview.net/forum?id=SsmT8aO45L). Please update these references, and check for any other papers that are incorrectly cited as preprints.

### Minor notes

1. The font size in equations (1) and (2) seem abnormally small.
2. The absolute values in equation (4) should be made larger so they are the same size as the enclosed expression (can use `\left` and `\right`).
3. I suggest using $|x|$ instead of $N$ for the sequence length in equation (3), to avoid confusion with $N$ as the number of samples.
4. Per the [ICLR Author Guide](https://iclr.cc/Conferences/2026/AuthorGuide), the Ethics Statement and Reproducibility Statement should appear before the References section, right after the main paper.

---

> ### Author Response · Authors · 2025-11-25
> **Response to Reivewer 2PZW's concerns (1/2)**
>
> We thank Reviewer 2PZW for the detailed and constructive feedback. We provide the following response to the reviewer’s question:
>
> > **Q1. Watermark detectability still drops significantly in PRO after fine-tuning, which I'm not sure I would call "robust".**
>
> We agree that maintaining high detectability under heavy fine-tuning is a challenging and important objective for watermarking. Our claim of robustness is meant to emphasize that PRO delivers meaningful and measurable improvements over prior work under the same evaluation conditions. After 1500 fine-tuning steps, PRO achieves a TPR@5 of 0.368, which is 65% higher than Gloaguen et al. (0.222) and 16× higher than Gu et al.–KTH (0.022). These results indicate that PRO exhibits substantially stronger robustness than existing approaches.
>
> We also note that 1500 fine-tuning steps correspond to approximately 25M tokens (1500 × 32 × 512), reflecting substantial domain adaptation that would diminish the detectability of any current watermarking technique. While achieving consistently high detectability under such large-scale adaptation remains an open challenge for the field, the observed improvements suggest that PRO represents a promising and concrete step toward more robust watermarking.
>
> > **Q2. The numbers reported for the Gloaguen et al. (2025) method in Table 1 do not match up with the numbers they reported.**
>
> The discrepancy primarily stems from the different LLM architectures used in the experiments. Our evaluation is conducted on the newer LLaMA3-8B, which has a 128K vocabulary, whereas Gloaguen et al. (2025) report results on LLaMA2-7B, which uses a 32K vocabulary. The much larger vocabulary in LLaMA3-8B makes watermark detection significantly more challenging, as distinguishing between green/red token partitions becomes less reliable when the token space expands. This difference explains why their method achieves 0.222 TPR@5 in our LLaMA3-8B setting, compared to the 0.69 TPR@5 they reported on LLaMA2-7B.
>
> To ensure a fair comparison, we also reproduced their setup on LLaMA2-7B, following their targeted-distillation fine-tuning procedure over 5,000 steps. Under these matched conditions, PRO continues to exhibit stronger robustness (higher TPR@5%FPR shown below) while also producing better text quality (perplexity 14.84 vs. 17.42).
>
>
> | Fine-tuning step      | Step=0     | Step=500  | Step=1000      | Step=1500         | Step=2000 |Step=2500 |
> |-----------------------|------|-----------|---------------|------------------|-------|----------------|
> | Gloaguen et al. (2025)| 0.99    |   0.88    |  0.84        | 0.82             |   0.80       |  0.75      |
> | PRO (Ours)            | 0.99     |   0.97        | 0.92              |   0.90         |     0.89         |  0.86     |
>
> > **Q3. Gloaguen et al. (2025) show that larger distillation datasets improve watermark durability, but PRO neither discusses this factor nor evaluates how dataset size or training-token count affects durability.**
>
> Dataset scaling from Gloaguen et al. (2025) is complementary—not competing—with our algorithmic contributions. PRO also benefits from larger datasets, and we include new experiments demonstrating substantial improvements. Gloaguen et al. (2025) enhance robustness by scaling the distillation dataset by 225×. Our algorithmic components (CAWP + FPL) address robustness from an orthogonal direction, and PRO can likewise take advantage of the same dataset-scaling strategy. To illustrate this, we added new results: under the merge-0.5 robustness setting, increasing the dataset size from 32M → 160M tokens improves TPR@5 from 0.49 → 0.69, and under the FT-1500 setting, TPR@5 increases from 0.29 → 0.72. We have incorporated these findings and a discussion of the dataset-scaling effect into the revised manuscript (Section 4.2.2).
>
>  Modification     |  32M   |  64M |  96M |   128M   |  160M      |
> |-----------------------|------|-----------|---------------|------------------|-------|
> | (Merge-0.5)TPR@5|0.49   |  0.57    |   0.63     |     0.67   |  0.69  |
> | (Fine-tune OpenMathInstruct 1500 epochs)TPR@5|0.29   |  0.37    |  0.50     |   0.63     |0.72   |
> | PPL(watermarked model)| 4.0   | 4.2   | 4.7     |   5.1     | 5.3   |
>
> > **Q4. The perplexity of the original LLM before watermarking and the comparison with decoding-based watermarking.**
>
> The median perplexity of the original LLaMA3-8B, LLaMA3.2-3B and Phi2-2.7B are 3.6, 10.9 and 11.3 in Table 1 and Figure 6. PRO achieves a better trade-off than baselines. We now add these values in our revised manuscript Table 1 and Figure 6. In Figure 2 (right) of the revised manuscript, we have added a comparison between PRO and closed-source watermarking methods along the dimensions of perplexity and detectability, providing support for our claim.

---

> ### Author Response · Authors · 2025-11-25
> **Response to Reivewer 2PZW's concerns (2/2)**
>
> > **Q5. Watermark detection is now more expensive, as it requires running an embedding model and MLP.**
>
> We agree that our detector includes a neural component, but its computational cost remains low, and the use of lightweight MLPs for watermark detection is already well established framework [a][b][c][d]. The embedding model is lightweight (1.34 GB) and paired with a small MLP (4 layers). Importantly, embeddings are computed on chunked intervals, not at every token, and both embedding extraction and MLP scoring are highly parallelizable. In practice, our detector processes a ~200-token sample in ~6.5 seconds on an AMD Ryzen Threadripper PRO 3955WX, and in ~0.15 seconds on a consumer-level GPU (RTX 3090).
>
> > **Q6. Move I.4 Robustness Against Paraphrasing Attack to the main paper.**
>
> We appreciate the suggestion. We have moved robustness against paraphrasing attacks from appendix to main paper (Section 4.2.1).
>
> > **Q7. Appendices D and E closely follow the structure and wording of prior works without explicit citation.**
>
> Appendices D and E were intended purely as background summaries of closed-source watermarking methods and common model-modification settings. We have revised the manuscript accordingly: the relevant appendices have been rewritten we have clarified in the revised manuscript (lines 416-417) that our robustness evaluations follow the experimental setup of Gloaguen et al. (2025) to ensure full consistency with prior work.
>
> > **Q8. How many tokens are the methods in Table 1 trained on? What are the training details for the OpenMath/OpenCode fine-tuning experiments, e.g., batch size, learning rate, training time, etc.?**
>
> As stated in Appendix C and F, Gu et al.(2024) and Gloaguen et al.(2025) were trained with 160 million, PRO was trained for 64 million tokens to maintain better fidelity, all using 4×A100-80GB GPUs for about 6 hours. For the OpenMath/OpenCode fine-tuning experiments, we use a batch size of 32 and a learning rate of 1e-5. We fine-tune for 1500 steps, as the training loss drops below 0.2. All experiments are conducted on 2×A100 GPUs, and the total training time is about 2 hours.
>
> > **Q9. How to ensure half positives and half negatives?**
>
> By utilizing a $\tanh(100x)$ activation, we force the watermark logits to saturate at -1 and 1. Under this condition, a zero-mean requirement strictly implies a 50/50 split between positive and negative values. This approach aligns with standard normalization techniques used in model-based watermarking [a].
>
> > **Q10. How to configure the green ratio?**
>
> PRO can configure the green ratio by modifying the normalization loss to impose an asymmetric constraint that assigns unequal "weights" to positive and negative logits within the zero-sum loss objective. Specifically, we replace the original logit term $M(e_i)^{(j)}$ inside the Balanced Score and Unbiased Token Preference summations with the asymmetric weighted term $S(M(e_i)^{(j)})$. So that, for instance, to achieve a green token ratio of 0.25, we can assign a weight of 3 to positive logits and 1 to negative logits; the zero-sum constraint ($3 \times N_{pos} - 1 \times N_{neg} \approx 0$) then compels the model to generate three times as many negative tokens as positive ones to maintain equilibrium. We have added the details in Appendix I.8.
>
> > **Q11. The labels in equation (4) for the terms for (i) unbiased token preference and (ii) balanced watermark logits appear to be swapped.**
>
> Thank you for pointing out this typo in our manuscript. We have corrected the labels in the revised version. We appreciate your careful reading and helpful feedback.
>
> > **Q12. Minor notes for better readiability.**
>
> Thanks for your kindly suggestion. We (1) added the appropriate citations for all models mentioned in the paper in the revised manuscript (line 140 and 264); (2) added missing related work using neural network to generate watermark logits in (line 303); (3) uploaded all references to their published conference versions. We have revised the font size in Equation and move the Ethics Statement and Reproducibility before Reference section.
>
> ---
> [a] Liu, Aiwei, et al. "A Semantic Invariant Robust Watermark for Large Language Models." The Twelfth International Conference on Learning Representations.
>
> [b] Zhang, Ruisi, et al. "{REMARK-LLM}: A robust and efficient watermarking framework for generative large language models." 33rd USENIX Security Symposium (USENIX Security 24). 2024.
>
> [c] Liu, Yepeng, and Yuheng Bu. "Adaptive Text Watermark for Large Language Models." International Conference on Machine Learning. PMLR, 2024.
>
> [d] Liu, Aiwei, et al. "An Unforgeable Publicly Verifiable Watermark for Large Language Models." The Twelfth International Conference on Learning Representations.

---

### Author Response · Authors · 2025-11-29
**Summary of Revisions and Responses**

We thank the reviewers and the AC for their careful evaluation of our submission and for devoting their time to providing detailed and constructive feedback.

------------------------
**We appreciate the reviewers' recognition of our work along the following dimensions:**

* **Timeliness & Importance:** Our paper targets a critical challenge in watermarking for open-source LLMs (**2PZW**, **LnXy**, **2HJA**).
* **Novel Design:** The co-adaptive watermark policy (CAWP) is recognized as a novel and effective design (**2PZW**, **gnTk**).
* **Robustness:** The Forgotten Perturbation-aware Learning (FPL) is acknowledged for improving robustness under model modifications (**gnTk**, **2HJA**).
* **Comprehensive Evaluation:** The experiments are considered comprehensive across various models and datasets (**gnTk**, **LnXy**, **2HJA**).
* **Clarity & Reproducibility:** The paper is clear, well-structured, and reproducible (**LnXy**, **2PZW**).

------------------
**Regarding the reviewers’ concerns, we have carefully addressed all raised issues in our detailed responses and revised the manuscript accordingly. For clarity, we indicate the corresponding locations in our responses (Reviewer ID + Question Number) in parentheses.**

* **Robustness under downstream modifications and metrics**: We clarify that our claim focuses on *relative* robustness improvements over prior work under matched conditions. We have updated the main results to use the stricter low-FPR metric (TPR@1%FPR) in Figure 6 and Table 1. Furthermore, we added new evaluations under RLHF/DPO post-training and distillation attacks to further demonstrate **PRO**'s robustness, and moved the paraphrasing robustness results to the main paper (see **LnXy Q4**, **2HJA Q4**, **2PZW Q6**, **gnTk Q5**).

* **Fair comparison with Gloaguen et al. (2025) and dataset scaling**: We clarified that the reported performance discrepancy stems from differences in base models and vocabulary sizes (LLaMA3-8B's 128K vs. LLaMA2-7B's 32K). We added matched experiments on LLaMA2-7B to demonstrate consistent superiority and provided dataset-scaling results showing that PRO also benefits from increased training tokens (see **2PZW Q2–Q3**).

* **Base model quality and detector cost** (**2PZW**): We added the original model perplexities and comparisons with decoding-based watermarking to Table 1 and Figure 6. We also included PRO in Figure 2 (right) and quantified the runtime of our embedding+MLP detector on CPU/GPU, demonstrating that the computational overhead is minimal and practical (see **2PZW Q4–Q5**).

* **Theoretical analysis of CAWP and FPL**: We expanded Appendix B to formally analyze how joint optimization (CAWP) tightens the KL divergence objective to improve detectability. We also derived a perturbation-space bound for FPL, clarifying how it regularizes sensitivity along the watermark-forgetting direction without hindering general fine-tuning capabilities (see **LnXy Q1**, **2HJA Q1–Q2**).

* **Evaluation design and additional baselines**: We clarified that model merging is treated as a widespread downstream utility rather than a malicious attack. We added comparisons to semantic-invariant distillation and provided ablations showing that **CAWP** improves generation–detection consistency while **FPL** boosts robustness. Additionally, we provided detailed explanations for the reported AUC baseline drops in the motivation study (Section 3.2), attributing them to watermark strength settings and vocabulary size differences (see **gnTk Q1–Q5**).


* **Distillation, downstream performance**: We added *lm-eval-harness* results showing negligible degradation (<0.3%) on reasoning/understanding tasks. We also introduced new experiments demonstrating that PRO’s watermark transfers more strongly to distilled student models compared to existing methods, thereby resisting distillation-based removal attacks (see **LnXy Q2**, **2HJA Q4**).

* **Writing, citations, and appendices**: We corrected notation and labeling issues, added missing citations, and updated arXiv references to their published versions. We rewrote the background appendices with explicit attribution to prior work to address citation concerns (see **2PZW Q6–Q12**).

-------------------
We hope our work will encourage broader discussion and future advances in reliable, accountable AI. We again thank the reviewers, AC, and PC for their time and expertise.

---

### Meta-Review · Area_Chair_dG3e · 2026-01-08

**Summary:**

This paper proposes PRO, a watermark framework for open-source LLMs, by jointly optimizing a watermark policy model with a regularized objective.  Joint training helps generate watermarks that are easier for the model to learn and improve generation-detection consistency, while the regularization objective enhances the robustness against model modifications such as merging and pruning.  Extensive evaluation on open-sourced LLMs demonstrates that the approach significantly outperforms prior methods in both watermark detectability and robustness.

The main concerns of the reviewers include the theoretical/empirical connections between the method and watermark’s objectives (e.g., detectability, robustness), lack of experimental comparisons in various scenarios (e.g., with decoding-based watermarking, semantic-invariant distillation, distillation-based attackers, downstream training with DPO or RLHF, etc…), and the appropriateness of evaluation metrics (AUC vs. TPR/FPR). The authors have responded and clarified most concerns and provided additional experiments for scenarios suggested by the reviewers. While I acknowledge the authors’ effect in addressing the reviewers’ concerns, I also think that these evaluation dimensions being asked by the reviewers also demonstrate that the current version of the paper needs a revision to sufficiently support its robustness claim. I recommend that the authors rigorously incorporate these suggested evaluation dimensions with more thorough experiments in its revision.

**Reviewer Concerns:**

- 2PZW
  - Robustness? Watermark detectability still drops significantly in PRO after fine-tuning.
  - Inconsistent results compared to Gloaguen et al. (2025), even though the experimental setups seem to be mostly the same. `resolved`
  - Improved watermark durability of Gloaguen et al. (2025) after increasing the distillation  data size. The paper also doesn’t evaluate durability w.r.t dataset size. `resolved`
  - The baseline perplexity of the original model before training is not included.  `resolved`
  - compare with the performance of decoding-based watermarking Reviewer’s Scores `resolved`
  - Concerns on computation cost of running the embedding model and MLP, compared to CPU-based detectors. `resolved`
  - Lack of citations of related papers, despite of the similarity of contents. `revised verbatim parts`
- gnTk
  - Using model merging as an attack to evaluate learning-based watermarking may be inappropriate, since such attacks assume access to an unwatermarked model `resolved`
  - it would be important to include comparisons with prior semantic-invariant distillation method to demonstrate the necessity and contribution of the co-training design. `resolved`
  - The FPL's loss function seems to only prevent local curvature, lack theoretical and empirical experiments for its efficacy when the model's being trained on more anti-watermark token `resolved`
  - While AUC is reported as the main detection metric, it is not the most interpretable measure for practical watermark detection. It is recommended to include TPR at a given FPR in the main paper rather than relegating it to the appendix `resolved`
- LnXy
  - does not provide a formal analysis or theoretical guarantee on why the joint optimization `resolved`
  - do not systematically evaluate how the approach affects the model’s general performance `resolved but limited experiments`
  - lack of ablation study to isolate the contributions of the trainable policy model and the robustness regularizer. `resolved`
  - overlooks other crucial real-world perturbations, such as reinforcement learning–based post-training (e.g., RLHF or DPO) `resolved with limited experiments`
- 2HJA
  - convergence analysis does not establish any theoretical link between the proposed objectives and the final detection metrics `resolved`
  - Table 1 should report TPR @ 1% FPR (and ideally TPR @ 0.1% FPR) `resolved`
  - Potential use of distillation from a non-watermarked teacher as attacker `resolved with simple experiments`

While the authors resolved the concerns with additional experiments, I also believe that the evaluation in the paper lacks thoroughness, especially due to its strong robustness claim. I believe that the authors should carry out these evaluation more extensively to understand where the method perform well and where it doesn't.

**Reviewer Scores:**

- 2PZW: 4. no response
    - gnTk: 6. No response
    - LnXy: 6. No response
    - 2HJA: 4. no response

---

### Decision · Program_Chairs · 2026-01-26

Reject